# Evaluation of the performance of four chemical transport models in predicting the aerosol chemical composition in Europe in 2005

M. Prank[1], M. Sofiev[1], S. Tsyro[2], C. Hendriks[3], V.S. Semeena[2], X. Vazhappilly Francis[4], T. Butler[5], H. Denier van der Gon[3], R. Friedrich[6], J. Hendricks[7], X. Kong[4], M. Lawrence[5], M. Righi[7], Z. Samaras[8], R. Sausen[7], J. Kukkonen[1], R. Sokhi[4]

[1] Finnish Meteorological Institute, Helsinki, 00560, Finland
[2] met.no, Norwegian Meteorological Institute, Oslo, Norway
[3] TNO, Netherlands Organization for Applied Research, Utrecht, The Netherlands
[4] Centre for Atmospheric and Instrumentation Research (CAIR), University of Hertfordshire, Hatfield, UK
[5] Institute for Advanced Sustainability Studies, Potsdam, Germany
[6] IER, University of Stuttgart, Stuttgart, Germany
[7] Deutsches Zentrum für Luft- und Raumfahrt (DLR), Institut für Physik der Atmosphäre, Oberpfaffenhofen, Germany
[8] Aristotle University of Thessaloniki, Thessaloniki, Greece

*Correspondence to*: M. Prank (marje.prank@fmi.fi)

**Abstract.** Four regional chemistry transport models were applied to simulate the concentration and composition of particulate matter (PM) in Europe for 2005 with horizontal resolution ~20 km. The modelled concentrations were compared with the measurements of PM chemical composition by the EMEP monitoring network. All models systematically underestimated $PM_{10}$ and $PM_{2.5}$ by 10-60%, depending on the model and the season of the year, when the calculated dry PM mass was compared with the measurements. The average water content at laboratory conditions was estimated between 5 and 20% for $PM_{2.5}$ and between 10 and 25% for $PM_{10}$. For majority of the PM chemical components, the relative underestimation was smaller than that, exceptions being the carbonaceous particles and mineral dust. Some species, such as sea-salt and $NO_3^-$, were overpredicted by the models. There were notable differences between the models' predictions of the seasonal variations of PM, mainly attributable to different treatments or omission of some source categories and aerosol processes. Benzo(a)pyrene concentrations were overestimated by all the models over the whole year. The study stresses the importance of improving the models' skill in simulating mineral dust and carbonaceous compounds, necessity for high-quality emissions from wildland fires, as well as the need for an explicit consideration of aerosol water content in model-measurement comparison.

## 1 Introduction

Exposure to particulate air pollution has been estimated to be among the ten most significant risk factors for public health globally, and among the 15 most relevant for Europe (Lim et al., 2012), substantially increasing the risks of respiratory and heart diseases. Recently air pollution and especially the particulate matter were classified as carcinogenic by WHO (Loomis

et al., 2013). Substantial research efforts have been dedicated to assess the health relevance of specific aerosol chemical components, although results are still largely inconclusive (Stanek et al., 2011). Particulate matter has also been recognized as a strong climate forcer that influences the Earth's energy balance through direct radiative effects and cloud processes. Clouds and aerosols contribute the largest uncertainty to the radiative budget estimates (IPCC, 2013). Both aerosol radiative properties and its ability to serve as a cloud condensation nuclei depend critically on its composition. The above-mentioned aerosol effects make it important for the atmospheric chemistry and transport models to accurately assess not only the total PM amount but also the particle chemical composition, size spectra and other physical and chemical features.

A systematic underestimation of total PM (also called PM deficit) has been frequently reported in chemical transport modelling studies (Bessagnet et al., 2016; Im et al., 2014; Solazzo et al., 2012a; Stern et al., 2008). In many cases such underestimation is to be expected: owing to the high complexity and uncertainty of associated emission and formation processes, models often omit some components of atmospheric aerosols and therefore fail to reproduce the total PM budget (Kukkonen et al., 2012). Among the most uncertain components are secondary organic aerosols (SOA) and natural emissions (forest fire smoke and wind-blown or re-suspended dust), which are often omitted or reproduced with large uncertainties by the models. Numerous studies have stressed the importance of these components. Perez et al., (2008, 2012); Putaud et al., (2004b, 2010) and Querol et al., (2004) reported that the coarse fraction ($PM_{2.5-10}$) includes large contributions from mineral dust, particularly in southern Europe, while the fine fraction ($PM_{2.5}$) is dominated by carbonaceous particles and secondary inorganic aerosol (SIA) (Putaud et al., 2010). According to Belis et al. (2013), SOA makes up most of the organic carbon, especially in rural areas and during warm periods, whereas a noticeable contribution from biomass burning is visible during cold season indicating the impact of domestic heating. The modelling quality of these compounds suffers from the relatively small amount of available observational data for the carbonaceous and crustal compounds. Several dedicated efforts have recently been made in order to understand and quantify the errors in modelling of these components and adequately represent them in the total PM budget, e.g. the studies of Denier van der Gon et al., (2015) for residential combustion, Soares et al. (2015) for wildfire emission, Kim et al. (2014) for wind-blown dust, Arneth et al. (2008) for biogenic VOC emissions. Modelling studies of SOA formation include those by Bergström et al., (2012); Ots et al., (2016), and Shrivastava et al., (2011)..

A specific challenge of the model-measurement comparison for individual PM components is the difference in how PM composition is represented in the models and observations. The observations are available for specific molecules or ions ($Na^+$, $SO_4^{2-}$, $NH_4^+$, $NO_3^-$, $Ca^{2+}$, Al, Fe, etc.) and elemental and organic carbon (EC, OC), while in the models, the speciation of primary aerosols rather follows the emission categories, such as anthropogenic sources, wildland fires, sea salt or wind-blown dust, which all can include several of the measured components (see e.g. Kuenen et al. (2014) for anthropogenic emissions, Akagi et al. (2011) and Andreae and Merlet, (2001) for wildland fire smoke, Avila et al. (1998) for wind-blown dust). As a further complication, the PM speciation measurements do not resolve the whole PM mass. Observational studies of the PM mass closure (Putaud et al., 2004b; Sillanpää et al., 2006) have reported an unidentified fraction of fine PM reaching up to 20-30% of the gravimetrically determined aerosol mass, while it might be as large as 40% for coarse particles.

The explanations for this deficiency include possible artefacts in observations of semivolatile organic and inorganic components, unaccounted non-carbon atoms (e.g. O, H) in organic matter, uncertainties in estimating the concentration of the crustal particles, and most importantly aerosol-bound water. Although some model-measurement comparison studies (e.g. Tsyro, 2005) have stressed the importance for the models to take the aerosol water content into account, it is still not considered in the majority of the studies.

Within the TRANSPHORM project (www.transphorm.eu), four state-of-art chemical transport models (CTMs) – CMAQ, EMEP/MSC-W, LOTOS-EUROS and SILAM - were applied to predict PM concentrations in Europe for 2005. In this paper we evaluate the ability of these models to reproduce the chemical composition and the total mass of $PM_{10}$ and $PM_{2.5}$ by comparing the model predictions with the measurements at the EMEP network (www.emep.int). The effect of the omission of certain PM components by the models is investigated. Attention is paid to the role of the most uncertain components, such as carbonaceous aerosols, mineral dust and wild-land fire emissions, as well as the role of aerosol-bound water in the PM observations. In addition to the individual models, the median of the 4-member multi-model ensemble is compared with the observations.

Majority of the multi-model inter-comparison studies for particulate matter have considered either only the total PM mass or just a few PM components (Hass et al., 2003; Im et al., 2014; Solazzo et al., 2012a), and some of them have been concentrated only on specific environmental conditions (e.g. Stern et al., 2008) or limited areas (Vautard et al., 2007). In the current study the model error regarding the PM simulation is characterized against available measurements of PM mass and composition in whole Europe during the different seasons. The most prominent areas for model improvement are identified.

## 2 Input data and participating models

### 2.1 European Emissions in 2005

A new anthropogenic emission inventory was compiled within the TRANSPHORM project, with substantial updates regarding the EU-wide transport activities. The baseline emission data contains the following substances: NOx, $SO_2$, NMVOC, $CH_4$, $NH_3$, CO, $PM_{10}$, $PM_{2.5}$, EC, B[a]P (benzo[a]pyrene), and particle number (Denier van der Gon et al., 2014). The natural emissions of biogenic VOCs and sea salt were calculated online by each model. The wild-land fire emissions were provided by the Integrated System for wild-land fires IS4FIRES v.1 (Sofiev et al., 2009) and were injected as primary particles to a homogeneous layer up to 1km above the surface. An exception was the SILAM model that calculates the wildfire emissions online, based on the IS4FIRES v.2 calibration (Soares et al., 2015) and vertical profiles of (Sofiev et al., 2012). Desert dust was included only through the lateral boundary conditions; no wind-blown dust was emitted inside the modelling domain.

## 2.2 Global boundary conditions

The inflow of PM and gases through the lateral boundaries was prescribed according to global simulations by two different models. The aerosol boundary conditions were generated by the EMAC (ECHAM/MESSy Atmospheric Chemistry, Jöckel et al., (2006)) global model including the aerosol sub-model MADE (Modal Aerosol Dynamics model for Europe, adapted for global applications, Lauer et al., (2005, 2007)). Boundary conditions for gas phase chemical species were provided by the global chemical transport model MATCH-MPIC (Model for Atmospheric CHemistry and Transport, Max Planck Institute for Chemistry version, Lawrence et al. (1999) and von Kuhlmann et al. (2003), Butler et al. (2012)). A detailed description of the models and the simulation setups can be found in Appendix A.

## 2.3 The regional models

The setups of the four participating models are summarized in Table 1. The detailed model descriptions are given in Appendix B.

The collected model output consists of hourly concentrations of each PM component, separately for fine ($PM_{2.5}$) and coarse ($PM_{2.5-10}$) fractions: $SO_4^{2-}$, $NO_3^-$, $NH_4^+$, EC, OC, SOA, sea salt, mineral dust, wild-land fire originated particulate matter, unspeciated other primary PM, and additionally also total $PM_{2.5}$ and $PM_{10}$ fields. While the primary anthropogenic PM, EC, secondary inorganic species and sea salt were computed by all models, other components were not always available (Table 2). For instance, OC was provided as a separate species only by EMEP and CMAQ models that included the secondary organic aerosol formation, while in the case of SILAM and LOTOS-EUROS primary OC was lumped with the rest of the anthropogenic primary PM. Due to very high uncertainties in the forest fire emission inventory, this component was left out of the total PM output of EMEP and LOTOS-EUROS, but was still provided as a separate field. In CMAQ the fine fraction of fire emitted PM was included in primary OA and the coarse fraction in unspeciated coarse primary PM.

Models also computed the concentration of benzo[a]pyrene (BaP), which was assumed to be an inert fine aerosol not participating in any chemical transformations and not affecting the total-PM budget due to its very low concentrations.

The ensemble median fields for total PM and each separate chemical component listed in Table 2 were computed from the hourly model data from the CTMs (hereinafter, median model). To reduce the influence of the components omitted in some of the models to the total PM, the median fields of the PM components were added up to form another dataset of total PM (hereinafter, medianComp model). When computing the median field for every component only those models are used which provided a valid field for that component, and thus the medianComp PM includes valid fields for all species computed by at least one model.

## 2.4 Observational data

The PM observations of the EMEP network were used for the model evaluation (Table 3). A detailed description of EMEP observations of PM and its components for 2005 is available in (Yttri et al., 2007). Table S1 shows the location and altitude

of all stations together with a list of observed species. EC/OC observations were available at four stations, which, along with data for a wide range of other species at these sites (Table 4) allowed a detailed evaluation of the PM composition along a transect from northern to southern Europe formed by these stations (Birkenes in Norway, Melpitz in Germany, Ispra in Italy and Montseny in Spain, Fig. S1).

In addition to the regular monitoring data, the EMEP 2002-2003 EC/OC campaign data are used for evaluation of the seasonality of the carbonaceous aerosols. The data were collected at 12 stations, one day per week from July 2002 to June 2003. One station in Portugal (Braganca, PTR0001R) was excluded from the comparison due extremely high modelled wild-land fire contribution which made that station not representative of average conditions – 2005 and 2003 were both record high wild-land fire years in Portugal, while being closer to average in rest of Europe. However, the 2002-2003 EMEP

intensive campaign ends in the beginning of July 2003 and thus does not cover the 2003 Portuguese fires which mostly took place in August.

## 2.5 Model measurement comparison

For the model-measurement comparison, the hourly model results were extracted at the station locations and averaged to the temporal resolution of the observations. The model data were converted to the observed quantities. The observed $Na^+$ was

assumed to originate only from sea salt, sea salt consisting 30.8% of sodium by dry weight. The part of the $Ca^{2+}$ observations not related to sea salt (nss-$Ca^{2+}$) was used to evaluate the modelled mineral aerosol. The sea salt related calcium was subtracted from the observations proportionally to observed $Na^+$ concentrations, sea salt including 1.2% of calcium by dry weight. Widely varying calcium contents have been reported for Saharan dust from different origin areas ranging from <5% to >15% (Avila et al., 1998; Formenti et al., 2011; Marconi et al., 2014; Putaud et al., 2004a). The calcium content of

anthropogenic emissions also varies between the sources, ranging from less than a percent for biomass burning (Akagi et al., 2011; Larson and Koenig, 1993) to ~30% for cement and lime production (Lee and Pacyna, 1999; van Loon et al., 2005). In the current study the modelled dust originating from the boundary conditions was assumed to come mainly from Sahara and was attributed 10% $Ca^{2+}$ content (Marconi et al., 2014). In addition, 3.5% $Ca^{2+}$ content was attributed to the mineral part of primary anthropogenic emissions. This value was chosen as it maximizes the correlation between the observed nss-$Ca^{2+}$ and

the model results. It stays well within the reported range for the anthropogenic emissions. The simulated nss-Ca concentrations were estimated as the sum of the 10% of dust concentrations plus 3.5% of the unspeciated other primary PM concentrations.

The OC to OM ratios have been reported to range from 1.2 to 1.6 for fresh anthropogenic emissions, while factors around 2 have been found for aged, secondary and oxygenated aerosol and particles originating from biomass burning (Aiken et al.,

2008; Turpin and Lim, 2001). Factor 1.6 was used in this study, analogously to (Bessagnet et al., 2014), however, this might be an underestimation for the EMEP stations, which are mostly located in rural areas and would thus be largely influenced by aged aerosols.

The aerosols emitted by wild-land fires also consist mainly of carbonaceous compounds. The fire emissions originated from IS4FIRES (Sofiev et al., 2009), which provides unspeciated $PM_{10}$ and $PM_{2.5}$ emissions. In SILAM, EMEP and LOTOS-EUROS the emitted PM was transported as a separate field of unspeciated particulates, while in CMAQ the fine fraction was included in primary OA and the coarse fraction in coarse primary PM. The fire OA in CMAQ cannot be distinguished from the anthropogenic OA, and fire EC was not included in that model. In the other models the fire PM has been further speciated as post-processing following Akagi et al., (2011) and Andreae and Merlet, (2001). On average these papers suggest roughly 5% EC and 50% OC content for fire emitted aerosol, the rest mainly consisting of non-carbon atoms in the organic compounds and some inorganics (up to 5%). The fire contribution to EC and OC has been calculated following this composition and added to the modelled EC and OC.

The models provided dry PM concentrations, which exclude the aerosol-bound water. In gravimetric sampling, which is the reference method for PM observations defined by the European Committee of Standardization, the filters are weighted in laboratory conditions of 20˚C and 50% relative humidity. While the deliquescence relative humidity of most pure inorganic salts present in aerosol is higher than 50% (Martin, 2000), it can be lower for mixed particles (Seinfeld and Pandis, 2006, chapter 10.2). Apart from that, hysteresis exists in the particle deliquescence-crystallization cycle. For some common aerosol components, such as ammonium sulphate and sodium chloride, the efflorescence humidity, at which the particle crystallizes and loses its water content, is below 50% (Martin, 2000). Therefore, if the particle has been exposed to a more humid outdoor environment, crystallization might not occur in the standard laboratory conditions, leaving some water bound to the particles on the filter. Based on the dry PM mass and speciation provided by the models, the aerosol thermodynamic model ISORROPIA2 (Fountoukis and Nenes, 2007) was applied to estimate the water content of the aerosol at the conditions where the filters were weighted (20°C, 50% relative humidity). ISORROPIA2 was run in the reverse mode, where the input quantities were the soluble inorganic components (SIA and sea salt, Ca) in the aerosol phase. Both stable and metastable states were computed, corresponding to the lower and upper branches of the deliquescence hysteresis loop, providing the lower and upper limits of the aerosol bound water amount.

The model results were evaluated in terms of bias, temporal and spatial correlations and the fraction of model values that are within a factor of 2 of the observations (FAC2).

## 3 Results of the model simulations

### 3.1 $PM_{2.5}$ and $PM_{10}$ concentrations in 2005

The annual mean $PM_{2.5}$ and $PM_{10}$ fields are presented in Figure 1 and Figure 2, for the individual models and the ensemble median. All models predict generally similar patterns of the near-surface concentrations for both $PM_{2.5}$ and $PM_{10}$ although there are significant quantitative differences between the models' predictions. For $PM_{2.5}$, the highest concentrations are in densely populated areas such as Benelux and Po Valley, which reflects the large contribution of anthropogenic sources. The $PM_{2.5}$ concentrations are lower over the open sea, whereas all models predict high $PM_{10}$ concentrations at marine areas due

to coarse sea salt contribution. However, large differences are visible in absolute $PM_{10}$ concentrations over sea, reflecting the differences between the sea salt emission algorithms. For example, the $PM_{10}$ level predicted by the EMEP model over sea is up to 4 times higher than that of LOTOS-EUROS, whereas SILAM predicts a considerable south to north decrease in the marine $PM_{10}$ concentrations due to the strong temperature dependence of its sea salt emissions. The LOTOS-EUROS

predictions did not include desert dust and wildland fire smoke, which explains the low values of both PM fractions close to the southern border of the domain.

The MedianComp model that sums up the ensemble medians of all the PM components and thus fully includes the wildfire emissions, desert dust and secondary organics, shows higher PM concentrations than the median model in various areas. The difference between the MedianComp and median models in the Central Europe is mainly due to SOA. $PM_{10}$ in the southern

part of the domain is influenced by the dust inflow from Sahara, while the fire impact is visible in Portugal. In CMAQ the dust and fire contributions are very low, and LOTOS-EUROS does not have them at all, so the median total PM is based on half of the models with zero or very low dust concentration. MedianComp is based only on the valid dust fields of SILAM and EMEP and thus includes noticeably higher dust contribution.

Figures S2 and S3 show the spatial patterns of model bias for $PM_{2.5}$ and $PM_{10}$ with regard to the EMEP network. The

15 individual models and the ensemble median underestimate both $PM_{2.5}$ and $PM_{10}$ concentrations quite homogeneously in space. The only station, where the models noticeably overestimate the PM concentrations, is located on the Schauinsland Mountain in the Black Forest, with an elevation above 1200 m. About 10 km from the station and about 1 km below it is Freiburg city. The overestimation occurs in winter (see the monthly average timeseries on Fig. S4), when the site is actually in the clear air above the low winter time boundary layer, while in the models, both the city and the station are covered with

20 one uniformly mixed grid cell. In summer, when the site is located within the boundary layer, the PM concentration there is mostly underestimated.

As seen from Figure 3, all models report stronger seasonal variations in total PM than is observed. The models report highest concentrations in autumn or winter, while the observations peak in spring. There are also noticeable differences between the models. In SILAM and LOTOS-EUROS the $PM_{2.5}$ concentration is noticeably lower in summer, while in CMAQ the autumn

concentrations are substantially higher than during the other seasons. EMEP predictions show very small seasonal variations for both $PM_{2.5}$ and $PM_{10.}$ The different anthropogenic emission seasonalities applied in the models (Table 1) explain part of the differences in Figure 3. However, omitting the secondary organic aerosol (SOA) is probably the main explanation for the exaggerated $PM_{2.5}$ summer minimums calculated by the LOTOS-EUROS and SILAM models. SOA is present in larger quantities in summer due to biogenic emissions of semivolatile organic compounds.

The model skill scores for $PM_{10}$ and $PM_{2.5}$ in winter and summer are presented in Table 5. The fraction of model values that are within a factor of 2 from the observations is larger in winter than in summer for all models, except EMEP. The temporal correlation of daily concentrations tends to be higher in winter, with the exception of CMAQ that has the lowest winter time correlations among the models. The models' ability to reproduce the average seasonal concentration patterns differs between finer and coarser particles – spatial correlation of $PM_{10}$ is higher in summer, while $PM_{2.5}$ correlates better in winter for most

of the models. Low summer-time correlations of LOTOS-EUROS result from large underestimations in Spanish stations due to missing Saharan dust. The summertime worse scores are probably due to the highly uncertain components that dominate the summer aerosol - wind-blown dust, wild-land fires, and biogenic secondary organic aerosols. The only score that is better in summer then in winter is the spatial correlation for $PM_{10}$. In summer, the $PM_{10}$ pattern over Europe is formed by the

inflow of Saharan dust and wild-land fires in Portugal and Spain which creates a strong south to north gradient. This gradient is reproduced by the models, although with smaller magnitude. As the species contributing to this summertime south-north gradient are desert dust and wild-land fires, which by nature are episodic and hard to model, the temporal correlation and factor-two agreement are still generally lower and bias is larger in summer. In winter, the particulate matter is dominated by the anthropogenic emissions, forming a more complex pattern, and thus the spatial correlation is worse.

As seen from Table 5, while the bias of the ensemble median follows the mean bias of the models, the temporal and spatial correlations exhibit more complicated relations. In winter, the ensemble median shows the overall best temporal correlation for both $PM_{2.5}$ and $PM_{10}$, while in summer EMEP performs better. The spatial correlations of SILAM or EMEP models usually slightly exceed that of the median model.

The medianComp fully includes SOA, desert dust, and fire-induced PM. As the contributions of those components are more

important in summer, the difference between the median and medianComp is largest in summer, and small in winter (Table 5). MedianComp thus shows a noticeably smaller summer-time bias than the median model for both $PM_{10}$ and $PM_{2.5}$. For $PM_{10}$ the medianComp outperforms the median model in summer in all quality scores, while for $PM_{2.5}$ its spatial and temporal correlations are worse. This indicates that accounting for desert dust, which is an important component in $PM_{10}$ and less so in $PM_{2.5}$, improves significantly the models' ability to reproduce the observed coarse PM patterns. It is worth pointing

out that the measurement network includes a large number of Spanish sites, where mineral dust is more important than for the rest of the modelling domain. The worsening of the summer time correlations of $PM_{2.5}$, on the other hand, indicates that improvements are necessary also for modelling the other components that were included only by few models, such as smoke from the vegetation fires and formation of secondary organic aerosols from the biogenic precursors.

The water contribution estimated with ISORROPIA2 based on the modelled aerosol composition is shown on Figure 3 with

light blue. The solid part indicates the stable water content (lower branch of the hysteresis cycle) and the striped part the metastable phase (the upper branch of the hysteresis cycle). In the stable case, the annual mean $PM_{2.5}$ water content, average over all EMEP stations, stayed between 4 and 9% depending on the model, and between 11 and 17% for $PM_{10}$. For $PM_{2.5}$, the models predicted annual average water content above 10% for only a few stations. For $PM_{10}$, CMAQ and EMEP predict majority of the stations to have less than 10% of water content, while LOTOS-EUROS and SILAM predict the majority to

be between 10 and 20%. Annual average water contents of more than 25% were predicted for some stations. The water content of $PM_{10}$ computed in the metastable mode was on average about twice higher (~25%); ~20% water content was predicted for $PM_{2.5}$.

As seen from Table 6, adding the aerosol-bound water reduces noticeably the model bias for both $PM_{10}$ and $PM_{2.5}$. For $PM_{2.5}$ the correlation coefficients are not much affected, while for $PM_{10}$, they are noticeably reduced. The factor-2 agreements

improve due to the bias reduction. The worsening correlations could be related to the models overestimating the sea salt concentrations that can lead to overestimation of the water content in $PM_{10}$, as sea salt is the most hydrophilic of the considered aerosol components.

## 3.2 PM composition in 2005

The ensemble median maps of the PM components are shown in Figures S5, S6, S7 and S8. In the Continental Europe the models predict the highest contribution from the summed secondary inorganic species, nitrate being most important in Central Europe and sulphate contributing mostly in Southern and Eastern regions. Sea salt concentrations are high over the marine areas and shores but decrease rapidly inland. Desert dust and wild-land fires can be the main contributors to aerosol in some areas, but their impact is spatially limited.

The models' performance in comparison to the measurements of the PM chemical components is shown in Table 7 (note different units ($\mu g$ N, S, C, Na, Ca / $m^3$); the model maps (Figures S5-8) that are plotted in full modelled species mass). The right columns of Figures S5 and S6 show the spatial spread of the model bias. $PM_{10}$ is underestimated slightly more than $PM_{2.5}$ by all models except EMEP, possibly due to the missing emissions of wind-blown dust, which mainly resides in the coarse fraction. Sodium and $NO_3^-$ are on average overestimated, whereas $NH_4^+$ and $SO_4^{2-}$ are underestimated but much less than total PM. The overestimation of $NO_3^-$ is most noticeable in the Central and Eastern Europe, whereas the western areas are predicted accurately and the northern ones are underestimated (Fig. S5). The carbonaceous aerosols and the mineral dust are underestimated more than the total PM.

Temporal correlation of the daily timeseries is usually lower for the specific components than for the $PM_{10}$ and $PM_{2.5}$, and same is true for FAC2. One possible reason for this is that summing up the non-correlated individual components smooths the gradients and reduces the penalty for slight mislocations of plumes. It cannot be ruled out that the lower correlation can in some cases be also due to higher observation errors. In particular, higher uncertainties are present in observations of mineral dust and carbonaceous species (Putaud et al., 2010, Annex 5; Sillanpää et al., 2006), but observation artefacts also influence the species with dynamic-equilibrium partitioning between particulate and gaseous phases, such as $NH_4NO_3$ (EMEP, 2001; Putaud et al., 2010, Annex 5). It also has to be noted that different pollutants are observed by different sets of stations in EMEP network, which might induce some extra variations to the average model scores.

The temporal correlations of the modelled carbonaceous compounds with their observed concentrations in $PM_{2.5}$ are among the highest for the PM components, and substantially lower for the observations of the same compounds in $PM_{10}$. The correlation coefficients are lowest for dust, but also below-average for $NO_3^-$. One can also notice a better agreement for the sum of $HNO_3$ and $NO_3^-$ than for $HNO_3$ or nitrates-only. Temporal correlation coefficients and factor-2 agreements are noticeably worse for $NH_3$ and $HNO_3$ compared with $NH_4$ and $NO_3$ aerosols or the gas-aerosol sums. The lower scores reflect the complexity of the gas-particle equilibrium between the $NH_4NO_3$ and $HNO_3$ and $NH_3.$ Another possible reason for higher scores for the summed gases and aerosols ($NH_3+NH_4$ and $NO_3+HNO_3$) than for $NH_3$, $HNO_3$ and nitrate aerosol separately is

the higher uncertainties in the observations of the latter (Chang et al., 2002; Schaap et al., 2011; EMEP, 2001; Putaud et al., 2010, Annex 5). Also the pulsed behaviour of the aerosol nitrate production in the PBL, recently described by (Curci et al., 2015), could be a reason for inaccuracies in modelling the nitrate aerosol. Conversely, for $NH_3 + NH_4^+$ the temporal correlation is lower than for $NH_4^+$ only, albeit the bias is smaller and FAC2 is better. Sulphate and $NH_4^+$ show very similar correlation values, as large fraction of $NH_4^+$ is present in the form of ammonium sulphate. The correlation for sulphates is higher than for $SO_2$, probably mainly due to the smoother features of the sulphate field – $SO_4^{2-}$ as a secondary pollutant is less affected by the local sources.

### 3.3 Secondary inorganic aerosols

The evaluation of the secondary inorganic aerosols (Figure 4) shows that the models reproduce relatively well the observed seasonal variation of SIA and its precursors. Moderate deviations exist: somewhat exaggerated seasonal cycle of $SO_2$ is shown by EMEP; CMAQ overestimates the autumn levels of both $NO_3+HNO_3$ and $NH_3+NH_4$, and predicts an autumn peak for all three SIA species; high autumn $NH_3+NH_4$ and $SO_4^{2-}$ are also predicted by SILAM and high autumn levels of $NO_3+HNO_3$ by EMEP. SILAM manifests strong over-estimation of sulphates in autumn – but no over-statement of $SO_2$. While the models reproduce well the summertime drop in the concentrations of $NH_4$ and $NO_3$, they tend to overestimate the autumn concentrations and while the observations show the highest concentrations in spring for all three SIA species, this is not reproduced by the models. This could be one of the reasons for the errors in the seasonal cycle of total PM.

The contribution of the gas phase $HNO_3$ and $NH_3$ to the sums of $NO_3+HNO_3$ and $NH_3+NH_4$, is shown on Figure 4 with dark shading. On annual average level the gas phase fraction is in both cases relatively well reproduced by most of the models, only SILAM underestimates $HNO_3$ and overestimates $NH_3$ contributions. However, the models do not reproduce well the seasonal variations in $HNO_3$ and $NH_3$ concentrations – EMEP and LOTOS-EUROS overestimate the seasonal variability of $HNO_3$, CMAQ strongly overestimates the autumn $NH_3$ concentrations and so does with smaller magnitude also SILAM. The seasonal variations of $NO_2$ are well reproduced, but all models apart from CMAQ overestimate $NO_2$ all the seasons Table S3 and Figure S9. This can be one of the reasons for the overestimation of the sum of $NO_3^-$ and $HNO_3$ seen on Figure 4.

### 3.4 Natural primary aerosols

For the sea-salt concentrations, EMEP and CMAQ predict higher levels than the other models and are also higher than observations in all seasons (Figure 5, left-hand column). However, the seasonal cycle is reproduced well. Conversely, LOTOS-EUROS, while being closest to the average annual level, underestimates the seasonal variations. SILAM is also close to the observations but seems to have an exaggerated temperature dependence of the sea salt emission as it overpredicts the summer and autumn concentrations while underestimating in winter. For all models the Na concentration for model-measurement comparison was computed from sea salt in $PM_{10}$, while the Na observations in EMEP network are made mostly in whole aerosol without size limits. As the models already overestimate Na concentration, including also the particles larger than 10 µm would lead to overestimation even larger than what is shown. However, comparing Na in $PM_{10}$

with Na in whole sea salt in SILAM (size range 0.01 to 30 μm), the changes are minor for majority of the EMEP stations that observed Na in 2005: below 5% for 65% of the stations, below 10% for 77%, and below 20% for all stations. The concentration changed more than 10% only in the stations located directly at seaside.

Only SILAM and EMEP modelled the transport of desert dust from the boundaries (mainly Sahara) as a separate tracer. A 10% $Ca^{2+}$ content was assumed for it (right panel of Figure 5, shaded part of the bars) and in addition a 3.5% $Ca^{2+}$ content was attributed to the mineral part of primary anthropogenic emissions (non-shaded part of the bars). The modelled contributions from these sources are about equal, except for winter when the models predict almost no dust from Sahara. The nss-$Ca^{2+}$ concentrations are substantially underestimated by the models for the whole year (EMEP model underestimated the nss-Ca by 75% and SILAM by 58%). Considering that the models omitted the wind-blown dust emissions inside the European modelling domain, this underestimation is not surprising. The seasonal patterns of the models differ from the observations, where the autumn concentrations are noticeably lower than the summer ones and close to the winter levels - the models rather suggest similar dust levels for most of the year, except for winter when the predicted concentrations are lower.

### 3.5 Carbonaceous aerosols

The available observations of the carbonaceous aerosols for 2005 point out a strong under-estimation of these components by the models (Figure 6, upper panels). The models underestimated the EC in $PM_{2.5}$ by ~20-60% and OC by 40-80% (Table 7). The models only provided the fine fraction of these compounds as separate tracers; the anthropogenic coarse mode emissions were included in the coarse unspeciated primary aerosol. The fire PM concentrations modelled by EMEP, LOTOS-EUROS and SILAM has been speciated as post-processing following Akagi et al., (2011) or Andreae and Merlet, (2001). On average these papers suggest roughly 5% EC and 50% OC content for fire emitted aerosol. The fire contribution to EC and OC calculated following this composition is shown on Figure 6 with darker shading.

The observations on the upper panels of Figure 6 are shown for OC and EC in both $PM_{2.5}$ (shaded part of the bars) and $PM_{10}$ (whole bars). The observations in $PM_{10}$ are about 20% higher than those in $PM_{2.5}$. The modelled fine EC and OC correlate substantially better with the observations in $PM_{2.5}$ than with those in $PM_{10}$ (Table 7). This agrees quite well with the emission estimates of Kuenen et al., (2014), according to which the anthropogenic emissions of coarse EC and OC are about 5 times lower than their fine mode ($PM_{2.5}$) emissions and also originate mostly from different sectors than the fine mode – coarse EC from large scale combustion and coarse OC from agriculture, while the most contributing sources of fine carbonaceous aerosol are residential combustion and traffic. As large part of OC is secondary and also resides in fine fraction, some extra sources are still necessary to explain the observed coarse OC. The contribution of the coarse mode to the OC concentrations is highest in summer and autumn and lowest in winter, consistent with origin from biological and agricultural sources.

SILAM only shows a small fire contribution to EC in spring and summer, while in EMEP and LOTOS-EUROS the contribution is larger and visible all year round. EMEP also predicts a noticeable fire contribution to OC for all seasons. For EMEP and LOTOS-EUROS, the fire contribution reduces the model bias for the carbonaceous species, while at the same time reducing the correlation with the measurements of EC and OC in $PM_{2.5}$ (Table S2). The SILAM EC prediction quality

does not noticeably change. The correlation with EC and OC observations in $PM_{10}$ in some cases rises when including the fire emitted part.

The models reproduce the observed seasonal variation in EC concentration, but all underestimate with varying magnitude. As elemental carbon emission data were the same for all models and no chemical transformations affect its concentrations in the atmosphere, the large differences of the average EC concentrations between the models are rather surprising. SILAM predicted the highest concentrations, being more than twice higher than CMAQ and EMEP in winter, the difference being smaller for the other seasons. A possible explanation is the considerably lower dry deposition of fine aerosols in SILAM (Kouznetsov and Sofiev, 2012). Different treatment of EC hygroscopicity and ageing, affecting the efficiency of its wet scavenging, could also contribute to differences in the model results. The relatively coarse vertical resolution near the surface is a plausible explanation of EMEP's underestimation of EC, especially in winter. Finally, the emissions of carbonaceous particles are likely to be underestimated during the cold seasons due to large uncertainties in the emission factors for the residential wood burning (Denier van der Gon et al., 2015).

For OC only CMAQ and EMEP results are included in the analyses, as OC was not available from LOTOS-EUROS and SILAM (these models did not calculate the secondary OC and lumped the primary anthropogenic OC into the primary PM emissions). The models did not reproduce the observed seasonal variations in OC concentration, which peak in winter and autumn - both models show quite flat seasonal profiles and if accounting for the wildfire emissions, EMEP even overestimates the summer concentrations.. The large underestimation in winter could be caused by missing emissions of domestic heating (Denier van der Gon et al., 2015), but also the SOA formation from anthropogenic aromatics could be underestimated. A rather large portion of semi-volatile organics is believed to be missing in current anthropogenic emission inventories of $PM_{2.5}$ and NMVOCs (Denier van der Gon et al., 2015; Donahue et al., 2006; Ots et al., 2016; Robinson et al., 2007). Cooking emissions have been pointed out as another missing source of organic aerosols (Fountoukis et al., 2015; Young et al., 2015).

The above analysis was based on only four stations that measured the carbonaceous compounds during 2005, which makes it uncertain. To better understand the results for carbonaceous compounds, we used OC/EC observations from the EMEP campaign in 2002/2003 (Simpson et al., 2007; Tsyro et al., 2007), when the carbonaceous aerosols in $PM_{10}$ were observed at 12 stations. Keeping in mind the inter-annual variability, some kind of indication of model biases can still be obtained from comparing the modelled seasonal average concentrations of EC and OC for 2005 with the seasonal averages of these observations, especially as the $PM_{10}$ concentrations observed during this campaign were underestimated by the models by about the same factor as the $PM_{10}$ observations of 2005. The comparison supports the previous conclusion: the modelled OC concentrations, and also those of EC at many sites, are substantially lower than the observations (Figure 6, lower row) and models completely miss the observed OC winter maximum.

## 3.6 Benzo(a)pyrene

All models of this study overestimated the Benzo(a)pyrene concentrations all year round (Figure 7) whereas the seasonal cycles are qualitatively similar to the observed. This is somewhat unexpected, as the models underestimate the concentrations of black carbon and the sources of these two pollutants significantly overlap. One possible reason for this can be a simplified approach taken by the models to simulate this species: BaP was assumed to be an inert fine aerosol not participating in chemical transformations and not partitioning to gas-phase. In more complex models the heterogeneous oxidation by ozone has been reported to efficiently reduce the BaP concentrations (Friedman and Selin, 2012; Matthias et al., 2009). It is also probable that some part of the over-estimation, especially in winter time when the oxidation is slower, may be attributed to the emissions.

## 3.7 PM composition in the four selected stations

The PM composition was evaluated at the four stations that provided more complete data on the chemical speciation of the PM concentrations (Fig. S1). All the modelled and observed species in Figure 8 are converted to total masses of the species in order to add up to total $PM_{2.5}$ or $PM_{10}$. OC is converted to total organic aerosol mass by multiplying with 1.6 and nss-$Ca^{2+}$ to mineral dust by multiplying with 10. Observed sea salt is taken as the sum of $Na^+$ and Cl. However, the modelled and observed species are not always directly comparable, e.g. some models include carbonaceous aerosol in primary anthropogenic PM or wildfire smoke and mineral dust in the primary unspeciated aerosol. OA comparison with observations is based on CMAQ and EMEP results only, since the other models include OC in other PM. Dust comparison is based on EMEP and SILAM.

As seen in Figure 8, for these stations the sum of measured PM components was up to ~20% lower than measured total $PM_{2.5}$ and $PM_{10}$. The water contribution estimated with ISORROPIA2 can be seen on Figure 8 in light blue. Adding the aerosol bound water in metastable state closes the gap between the observed total $PM_{10}$ and the sum of the individual components in almost all cases (in Montseny the $PM_{10}$ estimate based on nearby stations can be inaccurate). In Ispra and Birkenes the observed PM is exceeded, which could indicate that the aerosol on the filters is in crystallized state or be due to inaccuracies in other observed species.

At Melpitz the models are close to the observations for SIA and overestimate the sea-salt contribution. Carbonaceous part is underestimated, though accounting also for the wild-fire emissions (striped orange on Figure 8) brings EMEP very close to the OC observations in $PM_{2.5}$. The mineral dust transported from the boundaries (separately only in EMEP and SILAM) shows lower values than the observed dust concentration. EMEP is the only model, where the unspeciated part of the primary PM (PPMr) consists solely of mineral components, while in the other models it is mixed with either the primary organic aerosol or wild fire smoke. The sum of EMEP PPMr and desert dust is very close to the observation. However, here the observed total mineral dust concentration is estimated assuming 10% $Ca^{2+}$ fraction, which is an overestimation for majority of the anthropogenic emissions.

At Montseny all models overestimate $NO_3^-$, whereas $NH_4^+$ is overestimated by EMEP and LOTOS-EUROS and $SO_4^{2-}$ by SILAM. Considering that forest fire emissions also have substantial organic aerosol content, EMEP model is even overestimating the observed OA, while EC is overestimated by all models. Due to over-predicted $NO_3^-$, $PM_{2.5}$ is overestimated by EMEP and LOTOS-EUROS at this station. The modelled desert dust values are again substantially lower than the observed dust, while adding the PPMr concentration brings EMEP very close to the observation in $PM_{2.5}$, although still underestimating the mineral part of $PM_{10}$.

At Ispra, the major contributor to the observed PM is organic aerosol, while the models show a few times lower values. Elemental carbon is also somewhat underestimated. However, Yttri et al. (2007) warn against possible errors in the observations of carbonaceous aerosols at that site for 2005, especially in the case of $PM_{10}$. CMAQ also underestimates all SIA in Ispra and all models miss some $SO_4^{2-}$, while fine $NO_3^-$ is overestimated by LOTOS-EUROS and SILAM. Sea salt and dust cannot be evaluated in Ispra, as no $Na^+$ or $Ca^{2+}$ observations were available in 2005. This also makes the estimation of water content in the observed PM inaccurate.

At Birkenes all models but LOTOS-EUROS overestimate the measured $PM_{10}$. $PM_{2.5}$ is not shown, as the SIA, $Ca^{2+}$ and $Na^+$ observations were not available separately for fine and coarse aerosol. As these species were measured in total aerosol, they might partly also originate from larger particles than $PM_{10}$. Elemental carbon concentrations are somewhat overestimated by CMAQ and SILAM. EMEP overestimates the organic aerosol. All models overestimate the sea salt contribution in $PM_{10}$ by factor of 2-3, leading to very high water uptake of the aerosol. Modelled desert dust alone is lower than the nss-$Ca^{2+}$ based observation while its sum with PPMr brings EMEP again very close to the observation.

All-in-all, overestimations of some components can bring the models very close or even over the observed PM levels, while still underestimating other components. The sea-salt concentrations are usually overestimated by all models – up to a factor of 2-4 – and this becomes important at the sites with a significant sea salt fraction in the mass budget. Sulphates are reproduced comparatively well with limited regional differences, probably driven by emission data quality. $NH_4^+$ is quite well reproduced by all models, except for CMAQ, which under-estimates it. For nitrates, the models showed varying degree of agreement. OA is mostly underestimated, while EMEP can also sometimes overpredict its concentration. Models underestimate high observed EC observations, while low concentrations can be overestimated. Mineral dust, which was taken only from global boundary conditions, is not enough to explain the observed nss-$Ca^{2+}$ concentrations. Adding it up with the mineral part of the anthropogenic PM brings EMEP model close to observations, at least for $PM_{2.5}$. However, EMEP still underestimates the mineral contribution to $PM_{10}$ in Montseny, which is the station most influenced by Saharan dust. The underestimation of nss-$Ca^{2+}$ is smaller in the north, further away from Sahara (Fig. S6, lowest right panel).

**4 Discussion**

In the following we consider the major reasons for discrepancies of the model-measurement comparisons described above.

**4.1 Uncertainties in the model evaluation**

The individual PM components are reproduced with about the same or lower quality as the total PM. The temporal correlation of the daily timeseries is usually lower for the specific components than for the total PM, and same is true for the FAC2 agreement. This could indicate compensating errors in the model parameterizations, but even without that the comparison for the sum of the non-correlating components would benefit from the averaging of the errors in the components. The considered models are found to underestimate the observed total $PM_{2.5}$ and $PM_{10}$ concentrations. However, not all individual PM components are equally underestimated: secondary inorganic species are reproduced quite accurately and sea salt is usually overestimated. This suggests large underestimations for carbonaceous and mineral aerosols, which is supported by the few available observations. However, the mismatch between the modelled and observed quantities leaves large uncertainties in evaluating how much exactly these aerosol components are underestimated in this study.

Wind-blown crustal aerosols have been pointed out as a potentially underestimated fraction of PM (Im et al., 2014) and substantial underestimation is found also strongly indicated by this study. The fraction of calcium observations not related to sea salt was used to evaluate the mineral dust concentration in this study. However, the evaluation of the wind-blown dust against non-sea-salt calcium observations is highly uncertain. Various options exist for deriving the total mineral dust concentration from observations of e.g. aluminium or non-sea-salt fraction of calcium (nss-$Ca^{2+}$) (Marconi et al., 2014; Putaud et al., 2004b), but fractions of these vary among different minerals and dust source areas (Avila et al., 1998; Formenti et al., 2011). Putaud et al. (2010) provided various formulas for estimating the mineral dust concentration from several related tracers, such as Si, Al and Fe and nss-$Ca^{2+}$. They estimated that the uncertainty of deriving mineral dust concentration from observations can reach ±150%. Observations of Si, Al and Fe were available only in Montseny station. The location of Montseny station about 30 km from the Mediterranean coast, at 700m height from sea level exposes it to Saharan dust episodes (the high dust contribution there is visible on Figure 8) and thus allows for evaluating the nss-$Ca^{2+}$ as a desert dust tracer. The nss-$Ca^{2+}$ concentrations there correlate well (correlation coefficient above 0.9) with the observations of the other mineral dust tracers, and the dust concentration obtained by assuming 10% $Ca^{2+}$ content is not far from the estimates provided by the most detailed formulas presented in the Annex 5 of Putaud et al., (2010).

However, the wind-blown crustal emissions are not the only source of mineral aerosols. Generally, about half of primary fine anthropogenic aerosol emission consists of carbonaceous components (Kuenen et al., 2014), while the rest is mainly associated with mineral compounds. For coarse fractions, the carbon content is low; hence the bulk of mass consists of mineral components. Therefore, the unspeciated primary PM in the models has to also be included to the comparison with the nss-Ca observations. However, the variations of the calcium content are even wider there, ranging from less than a percent for biomass burning (Akagi et al., 2011; Larson and Koenig, 1993) to ~30% for cement and lime production (Lee

and Pacyna, 1999; van Loon et al., 2005). According to Lee and Pacyna, (1999), the emissions from coal combustion include 2% of $Ca^{2+}$ and steel and iron production emissions 0.7-3.6%. The $Ca^{2+}$ content in the top soil layer, influencing the dust emissions form agricultural activities, but also the dust suspended by wind and traffic, stays in Europe below 5% and below 1% in the northern areas (van Loon et al., 2005). Although the 3.5% $Ca^{2+}$ content used in this study for the anthropogenic

mineral aerosol is well within these limits, good model-measurement agreement cannot be expected due to these large variations. The uncertainty in aerosol Ca content can be expected to be a few times. However, with the 10 and 3.5% Ca content, the EMEP model underestimated the nss-Ca by 75% and SILAM by 58%, so even assuming twice the calcium content, the nss-calcium concentrations would still be underestimated by the models.

In 2005, the wild-land fires took place in a comparatively small part of the domain and affected noticeably only a few

stations in Spain and Portugal. However, the very strong emission within short time had a significant impact on PM concentrations even at annual level. Therefore, exclusion of this component from the computations results in strong under-estimation and poor correlation, both in space and in time. On the other hand, fire emission is arguably among the most uncertain input datasets (Soares et al., 2015) and requires careful treatment accounting for the strong diurnal variation of the fluxes, as well as the vertical injection profile. The fires emit wide spectrum of pollutants and the observations rarely

distinguish the fire-originated aerosol from the rest of atmospheric PM. Specific tracers of combustion of organic materials, such as levoglucosan, are occasionally measured, but their relation to the total emitted PM is not fixed. Also, wood burning is common in many other sources, such as domestic heating, which cannot be told apart from large scale fires. As a result, evaluating the modelled fire smoke becomes possible only for episodes with strong domination of fire-induced pollution .On the other hand, inaccurate representation of the fire emissions and their temporal and vertical profiles can result in a very

poor correlation with the measured concentrations. In EMEP and LOTOS-EUROS, using the IS4FIRES v1 emissions resulted in degradation of model scores for $PM_{2.5}$ and $PM_{10}$ and thus these models excluded the fire PM from their total $PM_{2.5}$ and $PM_{10}$ fields, the correlations for EC and OC also reduced when fire contribution was added (Table S2). In SILAM, a newer version of the emission data was used (IS4FIRES v2, Soares et al., 2015), together with dynamic emission vertical profiles (Sofiev et al., 2012), while in other models the IS4FIRES v1 emission data was spread evenly to the first 1000m.

Mainly due to the vertical profiles that release most of the smoke high aloft, the ground level concentrations of fire PM were substantially lower in SILAM and the fire PM did not negatively affect the model performance, demonstrating that the quality of the fire emission data is essential for simulating the particulate matter concentrations.

The spatial features of the compared data can also lead to uncertainties in model-measurement comparison. Regional models with grid-cell sizes of a few tens of kilometres are not designed to reproduce the concentration patterns with smaller spatial

scales, e.g. in the vicinity of strong sources, in urban conditions or mountainous areas. For instance, the study by (Im et al., 2014) found a stronger underestimation of PM in urban stations than in rural ones, which, apart from emission underestimation, could also be explained by the limited representative area of these stations. Also Vautard et al. (2007) found larger PM underestimation in the urban stations by the large scale models than by those with higher resolution. Even for stations of the EMEP network, whose locations have been carefully selected to represent the regional background

(EMEP, 2001), the effects of local topography and sources may still be noticeable. The models' performance was found to degrade in the higher stations - there is a strong negative correlation between the station altitude and the models' temporal correlation coefficients for both $PM_{10}$ and $PM_{2.5}$. The bad model performance is caused not only by the altitude difference between the station and the model grid cell average, but also other inhomogeneities, such as strong emission sources in the area. Indications of wintertime overestimation are visible for some of the high stations, but not all the high stations are located at extreme points of the terrain, such as mountain summits, and not all of them have strong emission sources in the immediate vicinity. Opposite problems arise for sites located in narrow valleys, where the models cell-mean altitude is higher than the station and the models overspread the pollution that in reality can be trapped in the valley.

Wang et al., (2014) and Samset et al., (2014) demonstrate that shorter EC lifetimes are necessary for reproducing the EC vertical profiles and low concentrations in remote regions. This result contradicts with the current model intercomparison, where SILAM was found to best reproduce the observed EC concentrations, and longer EC lifetime due to slower deposition in that model was assumed as the main reason for the model-to-model differences. Also the temporal correlation with 2005 observations and spatial correlation between the models 2005 average EC and EC observed during the 2002-2003 EMEP campaign is no worse for SILAM than it is for the other models, and hence there is no clear indication that the slower deposition would not be consistent with the surface EC observations in European scale. However, as indications were found of strong underestimation of EC emission, the slower deposition in SILAM is likely to be compensating for the missing emissions. Observations of vertical profiles and concentrations in more remote locations would be necessary for investigating this issue; unfortunately such were not available in Europe in 2005.

### 4.2 Seasonality of model skills, relation to PM composition

Seasonal variations of secondary aerosols result from a wide range of processes. Firstly, the emissions of precursors vary seasonally and some of these depend on meteorology. For instance, $NH_3$ emission depends strongly on the seasonality and type of agricultural activities, as well as on the temperature. Secondly, formation of secondary pollutants from precursor gases is controlled by multiple factors with strong seasonal cycles: the abundance of oxidants and water, ambient temperature and solar radiation, etc. Thirdly, gas-particle partitioning of semi-volatile species depends on temperature and relative humidity. There are significant differences in the treatment of these processes in the models, leading to substantial variations between the modelled seasonal cycles of the secondary aerosol concentrations. Resulting from these variations, the ability of the models to represent the observed $PM_{2.5}$ and $PM_{10}$ concentrations also varies seasonally and largely depends on the completeness of PM chemical composition in each specific model. For instance, the models that do not include SOA have larger bias in summer. Missing the contribution of the desert dust and wild-land fires also leads to negative bias and strongly reduces spatial correlation during summer time.

Especially for $NH_3$ the timing of the emissions as used in the models (fixed temporal profiles) can deviate substantially from real world emission timing which is largely controlled by meteorology (Backes et al., 2016; Hamaoui-Laguel et al., 2014; Hendriks et al., 2016). Meteorology also influences the total amount of emitted $NH_3$ but the strongest influence is on the

timing. The timing of agricultural activities, such as manure spreading has also a strong impact on the emissions (Hendriks et al., 2016). The inaccurate temporal emission profiles lead to models not reproducing the seasonal cycle of SIA (Figure 4) and PM.

The observed nss-Ca concentrations peak in spring and so do the modelled Saharan dust concentrations. Previous studies about Saharan dust confirm the emissions peaking at spring (Fiedler et al., 2013; Laurent et al., 2008). Additionally to Saharan emissions there are other reasons for elevated crustal aerosol concentration in spring, such as agricultural activities and vehicle caused erosion of roads - in colder regions where winter tires are used and the roads are sanded against slipperiness, high dust emissions occur when the road conditions get dry in spring. These emissions were not included in the model runs, which could be another reason why the models miss the spring peak in PM.

Another source of OC that has received very little attention is the primary biogenic particles, such as plant debris, fungal spores and pollen. While majority of these particles are larger than 10μm, the aerodynamic diameter of some common fungal spores is below 10μm and in some cases even below 2.5μm (Reponen et al., 2001), making them relevant to even $PM_{2.5}$. According to Hummel et al. (2014) and Winiwarter et al. (2009) the fungal spores could contribute noticeably to aerosol concentration in summer and autumn (up to a microgram m$^{-3}$ in long term average and even more during specific episodes).

The PM components mentioned above as the most uncertain and sometimes omitted in the models (wind-blown dust, wild-land fire smoke, biogenic primary and secondary particles), are all more common in summer time. The models mostly do underestimate PM by a larger fraction in summer. On the contrary, organic aerosol is underestimated by a larger fraction in winter. As noted by (Denier van der Gon et al., 2015; Lefebvre et al., 2016), the residential wood combustion emissions are severely underestimated in the current emission inventories and that would cause underestimation in carbonaceous particles during the cold seasons. According to Fountoukis et al., (2015) underestimation of the SOA formation rate in low light conditions could be another reason for the winter time OA underprediction.

### 4.3 Aerosol mass closure

Previous publications (Putaud et al., 2010; Sillanpää et al., 2006; Tsyro, 2005) have pointed out that a gap exists between the gravimetric total-PM observations and the sum of individual PM components (also seen in Figure 8). The main reason for this has been found to be aerosol-bound water contribution to the gravimetric observations, which can contribute ~20% of mass to annual average observations. Based on the modelled aerosol composition, the average water content at laboratory conditions was estimated roughly between 5 and 20% for $PM_{2.5}$ and between 10 and 25% for $PM_{10,}$ depending on whether the aerosol was assumed to be in stable or metastable state, the latter corresponding to situation, when the aerosol has been exposed to more humid conditions and crystallization has not occurred. Adding this contribution to the modelled PM reduced the model bias 25-70%, but also reduced both spatial and temporal correlations with the observations.

Several uncertainties exist estimating of the PM water content. Firstly, the water content depends on the outdoor humidity at the measurement location as well as the filter transportation and storage conditions, so it cannot be determined, whether the aerosol is in stable or metastable branch of the hysteresis cycle. Secondly, ISORROPIA2 computes the water content based

on the inorganic part of aerosol – SIA, sea salt, calcium; it does not take into account the water related to the hydrophilic part of the organic aerosol, which could also influence the water uptake of the inorganic species (Jing et al., 2015). Thirdly, the aerosols were assumed fully internally mixed, which lowers the deliquescence humidity compared to external mixtures and might lead to overestimation of water uptake. Overestimating hydrophilic compounds, such as sea salt can also lead to overestimation of the water content in PM. Also, in addition to the particle-bound water, the filters themselves can accumulate humidity and influence the measurement results (Brown et al., 2006). Taking into account all these uncertainties, the water content estimated based on the observed PM composition (Figure 8) assuming metastable state surprisingly well closes the budget for several stations (e.g. $PM_{10}$ and $PM_{2.5}$ in Melpitz and $PM_{2.5}$ in Montesny ).

Even when non-gravimetric measurement methods are used, they often include processing steps to obtain similar values to the gravimetric method, which is defined as the reference for PM measurements by European Committee of Standardization. The reason for these corrections is that a substantial fraction of secondary aerosols consists of components, such as ammonium nitrate and semivolatile organic species, whose partitioning between gaseous and particulate phase depends on the atmospheric conditions and concentrations of the compounds. Apart from water, also the semivolatile compounds can condense or evaporate during the measurement process. Loss of semivolatiles is an especially important issue for observation techniques that involve heated inlets, and dedicated methodologies have been developed to compensate for such losses and bring the results closer to the standard gravimetric observations (Alastuey et al., 2012; Charron et al., 2004; Hauck et al., 2004). However, such corrections implicitly introduce the particle water related offset also to observations that should by their design avoid it. As various applications using the $PM_{10}$ and $PM_{2.5}$ concentrations as an input (e.g. health impact assessment) are often calibrated using the total PM observations, using the model-produced dry PM masses will introduce a bias to the impact analysis.

## 5 Conclusions

The currently available chemical transport models commonly under-predict the PM mass concentrations, however the previous multi-model studies have not thoroughly investigated how this underprediction is reflected in the PM chemical composition. The current study was conducted to quantify the model deficiencies in terms of the aerosol chemical constituents, source categories, seasonal variations, and geographical distribution.

The aerosol predictions of four widely used chemical transport models (CMAQ, EMEP, LOTOS-EUROS and SILAM) were compared to the chemically-speciated PM observations by the EMEP monitoring network. All models showed comparable scores in reproducing the PM observations, generally underestimating the total PM mass by 10-60%, depending on the season of the year and the model. The PM components for which the modelling and monitoring experience is longer, such as nitrates, sulphates and ammonia were reproduced fairly well by all the models, whereas there were major underestimations for carbonaceous and mineral aerosols. The benzo(a)pyrene concentrations were overestimated by all models, probably owing to missing processes and inaccuracies in emission data.

The study highlighted the importance of the contribution of commonly omitted aerosol components, such as SOA, mineral dust and wildfire smoke. Neglecting the desert dust contribution to the PM budget substantially worsened the correlation of model predictions with PM observations in summer, which indicates that accounting for the inflow of Saharan dust is important in PM simulations, especially for southern Europe - for central and northern parts, agricultural and road dust are more important on an annual basis. The impact of wild-land fires was also significant in summer of 2005 in the western and southern parts of the domain. Including SOA in the modelled PM also substantially reduced the model bias in summer. Providing that all major PM components are included, the particle-bound water in gravimetric PM observations can explain a major fraction of the remaining bias.

The ensemble median showed better correlation with the observations than the individual models. However, the bias demonstrated by all models propagated also into the median results. This effect can be reduced by computing the median for each of the PM components separately with subsequent summation to the total-PM concentration. This procedure reduces the effect of the components that have been omitted by some of the models within the ensemble.

## Appendix A: Global models

### EMAC-MADE

EMAC is a numerical chemistry and climate simulation system describing tropospheric and stratospheric processes (Jöckel et al., 2006). It is based on the 5th generation European Centre HAMburg general circulation model (ECHAM5, Roeckner et al., 2006) and uses the Modular Earth Submodel System (MESSy) as an interface to couple various sub-models to the core model. Aerosol microphysics is simulated with the sub-model MADE (Lauer et al., 2005, 2007), which describes the aerosol population by means of three log-normal size modes, taking into account nucleation of new particles, condensation of sulphuric acid vapour and condensable organic compounds, and coagulation. MADE considers eight aerosol species: black carbon, particulate organic matter, sulphate, nitrate, ammonium, mineral dust, sea-salt, and aerosol water. Basic tropospheric gas-phase chemistry ($NO_x$-$HO_x$-$CH_4$-CO-$O_3$) and the sulphur cycle are simulated by the MECCA submodel (Sander et al., 2005). Additional processes include liquid phase chemistry (SCAV submodel, (Tost et al., 2006), gas/particle partitioning (Metzger et al., 2002), wet and dry deposition (SCAV and DRYDEP submodels, Kerkweg et al., 2006), aerosol activation during cloud formation (Abdul-Razzak and Ghan, 2000) and cloud microphysical processes simulated by the two-moment cloud scheme by Lohmann et al. (1999) and Lohmann, (2002). The EMAC-MADE model system has been evaluated by Lauer et al. (2005, 2007), Aquila et al. (2011) and Righi et al. (2013).

The emission setup considers biomass burning emission from the GFED dataset (van der Werf et al., 2010), anthropogenic emissions according to the RCP 8.5 scenario (Lamarque et al., 2010; van Vuuren et al., 2011) for the year 2005, and natural sources (volcanic $SO_2$, DMS, secondary organic aerosol). Wind-dependent number and mass emission fluxes are calculated on-line based on the parameterization of Guelle et al.,(2001) for sea salt and Balkanski et al. (2003) for desert dust. Dust is emitted in two log-normal modes with the size distribution parameters from (Dentener et al., 2006).

The EMAC simulations for this study were performed with a T42L19 resolution, i.e., with a horizontal spectral resolution with a triangular cut-off at great circle wave number 42, corresponding to a Gaussian grid of about 2.8° resolution and 19 vertical hybrid σ-pressure layers with the top layer centred at 10 hPa. The model dynamics were nudged to the operational analysis data of the European Centre for Medium-range Weather Forecasts (ECMWF).

## 5  MATCH-MPIC

Boundary conditions for gas phase chemical species were provided from the global chemical transport model MATCH-MPIC (Model for Atmospheric CHemistry and Transport, Max Planck Institute for Chemistry version, Lawrence et al. (1999) and von Kuhlmann et al. (2003)). The model was operated with input meteorological fields of the NCEP GFS (National Center for Environmental Prediction Global Forecast System). Tracer transport by advection, vertical diffusion and deep convection, as well as the tropospheric hydrological cycle (water vapour transport, cloud condensate formation and precipitation) are computed within the model. Chemical reactions of anthropogenic and biogenic NMVOCs are included, along with background tropospheric chemical reactions. More details on the simulations can be found in Butler et al. (2012).

## Appendix B: Regional models

## CMAQ

The Community Multi-scale Air Quality (CMAQ) modelling system applied in the study is the CMAQ version 4.7.1 with carbon bond chemical mechanism version 5 (Foley et al., 2010). The model grid was in Lambert conformal Projection (LCP) centred at $(54^o, 0^o)$ with standard parallel latitudes $30^o$ and $60^o$, respectively. CMAQ was applied on horizontal grid dimension with 18 km resolution. The study domain encompassed entire Europe with Atlantic Ocean as its western boundary. The CMAQ model consisted of 34 vertical layers extending from the surface up to ~20 km height. The meteorological inputs for the chemical transport model were generated from the meteorological modelling simulations of the Weather Research and Forecast (WRF) model version 3.2.1 (Skamarock et al., 2008). The WRF simulation was performed using 18km x 18km horizontal grid resolution with 52 vertical layers. The simulations used NOAA soil vegetation model applied as the land surface scheme, RRTMG as the long wave radiation scheme, Morrison scheme for microphysics parameterization, Grell and Devenyi scheme for cumulus parameterization, and YSU scheme for boundary layer parameterization. Meteorological initial and lateral boundary conditions were derived from the ECMWF analysis. In order to constrain the meteorological model towards the analyses a grid nudging technique was employed every 6 hours of WRF simulation. The results from WRF simulations were pre-processed for CMAQ using Meteorology-Chemistry Interface Process (MCIP) version 3.6 (Otte et al., 2005). In MCIP, 52 layers of the WRF model simulations were collapsed to 34 layers used in the CMAQ simulation.

The primary particulate matter such as $PM_{2.5}$, $PM_{10}$, elemental carbon, and sea salt as well as secondary inorganic aerosol species ($SO_4^{2-}$, $NO_3^-$ and $NH_4^+$) were included for the model comparison. The sea salt production in the marine boundary layer included the heterogeneous chemistry of sea salt aerosols (Spicer et al., 1998).

## EMEP/MSC-W

The EMEP/MSC-W model (Simpson et al., 2012) is a chemical transport model developed at the Meteorological Synthesizing Centre West of EMEP (http://www.emep.int), hosted by Norwegian Meteorological institute. At the same website, the model code (Open Source) and a suite of input data for a full year run are available. The model performance is regularly evaluated with EMEP routine monitoring and intensive measurement campaigns, as well as with other observational data (AirBase, satellite, sun-photometer, LIDAR measurements).

The calculations were performed using ECMWF-IFS meteorology, on $0.2° \times 0.2°$ grid, and the results were interpolated to the unified $0.3° \times 0.2°$ grid. The vertical distribution was resolved with 20 layers, reaching 100 hPa, with the lowest layer being approximately 90 m thick. Calculated concentrations were interpolated between the model layers to provide data at the requested levels, i.e. 100, 500, 1000, 3000 m), in addition the concentrations at a height of 3 m were derived from the results in the lowest layer for comparison with observations. The emission data, including forest fires, and boundary conditions were harmonized with the other participating models as described in sections 3.1 and 3.2 but the temporal emission profiles

followed (Simpson et al., 2012). The model included all main aerosol components from anthropogenic and natural sources, namely $SO_4^{2-}$, $NO_3^-$, $NH_4^+$, elemental and organic (both primary and secondary) carbon, sea salt and mineral dust (here only from the boundary conditions). $SO_4^{2-}$ is formed through $SO_2$ homogeneous and heterogeneous oxidation; $NO_3^-$ and $NH_4^+$ are calculated through aerosol-gas partitioning using thermodynamic equilibrium model MARS. In addition, the formation of

coarse $NO_3^-$ is included in a simplified way. Describing dry and wet deposition, the model treats separately fine and coarse aerosols.

## LOTOS-EUROS

In this study we used LOTOS-EUROS v1.8, a 3-D regional CTM that simulates air pollution in the lower troposphere (Schaap et al., 2008, Wichink Kruit et al., 2012). The calculations were performed with longitude–latitude $0.3° \times 0.2°$ grid.

The model vertical spans up to 3.5 km above sea level and consists of three dynamical layers: a mixing layer and two reservoir layers above it. The height of the mixing layer at each time and position is extracted from ECMWF meteorological data used to drive the model. The height of the reservoir layers is set to the difference between ceiling (3.5 km) and mixing layer height. Both layers are equally thick with a minimum of 50 m. If the mixing layer is near or above 3500 m high, the top of the model exceeds 3500 m. A surface layer with a fixed depth of 25 m is included to monitor ground-level concentrations.

Advection in all directions is handled with the monotonic advection scheme developed by Walcek, (2000). Gas phase chemistry is described using the TNO CBM-IV scheme (Schaap et al., 2009), which is a condensed version of the original scheme by Whitten et al. (1980). Hydrolysis of $N_2O_5$ is described following Schaap et al. (2004a). Aerosol chemistry is

represented with ISORROPIA2 (Fountoukis and Nenes, 2007). The pH dependent cloud chemistry scheme follows Banzhaf et al. (2012). Formation of coarse-mode nitrate is included in a dynamical approach (Wichink Kruit et al., 2012). Dry deposition for gases is modelled using the DEPAC3.11 module, which includes canopy compensation points for ammonia deposition (Van Zanten et al., 2010). Deposition of particles is represented following Zhang et al. (2001). Stomatal resistance

is described by the parameterization of (Emberson et al., 2000a, 2000b) and the aerodynamic resistance is calculated for all land use types separately. Wet deposition of trace gases and aerosols are treated using simple scavenging coefficients for gases (Schaap et al., 2004b) and particles (Simpson et al., 2003). Biogenic VOC emissions (Schaap et al., 2009) are derived from a dataset with the distributions of 115 tree species as obtained from Koeble and Seufert, (2001). Emissions of sea salt particulates (following Mårtensson et al., 2003; Monahan et al., 1986) are taken into account. The temporal variation of

anthropogenic emissions is represented by monthly, daily and hourly time factors for each source category (Builtjes et al., 2003). The model set-up used here does not contain secondary organic aerosol formation.

**SILAM**

The System for Integrated modeLling of Atmospheric coMposition (SILAM; http://silam.fmi.fi, Sofiev et al., 2015) is a global-to-meso-scale chemical transport model developed at FMI and used in research and operational applications related to

air quality and emergency. SILAM uses a transport algorithm based on the Eulerian advection scheme of (Sofiev et al., 2015), and the adaptive vertical diffusion algorithm of (Sofiev, 2002). The model includes a meteorological pre-processor for diagnosing the basic features of the boundary layer and the free troposphere (such as diffusivities, similarity scales, and latent and sensible heat fluxes) from meteorological fields provided by various meteorological models (Sofiev et al., 2010). For secondary inorganic aerosol formation, the updated chemistry scheme from DMAT model (Sofiev, 2000) was extended

with the coarse-nitrate formation. The dry deposition scheme is described in (Kouznetsov and Sofiev, 2012). Sea-salt was emitted according to (Sofiev et al., 2011), the size distribution being represented by 5 bins from 0.01 to 30 μm. Wild land fire emissions of IS4FIRES v.2 (Soares et al., 2015) were used.

The SILAM model has been extensively evaluated against air quality observations over Europe and the globe (Huijnen et al., 2010), http://www.gmes-atmoshpere.eu, http://www.myair.eu (Solazzo et al., 2012a, 2012b). The model has recently been

applied to evaluate the dispersion of primary $PM_{2.5}$ emissions across Europe and in more detail over Finland, and to assess the resulting adverse health impacts (Karvosenoja et al., 2011; Tainio et al., 2009, 2010).

For TRANSPHORM, the computations were made using meteorological fields from ECMWF operational forecasts from 2005. The computational grid covered the domain with spatial resolution of $0.3° \times 0.2°$, vertical grid consisting of 8 unevenly spaced layers stacked up to ~8km. The aerosol components included secondary inorganic species $SO_4^{2-}$, $NO_3^-$ and

$NH_4^+$; primary particulate matter $PM_{2.5}$ and $PM_{10}$, elemental carbon, dust, and sea salt.

## Acknowledgements

The study has been performed within the scope of EU the project FP7-ENV-2009-1-243406 (TRANSPHORM). Fire emission and SILAM development was supported by Academy of Finland projects APTA and ASTREX. DLR is grateful to DKRZ for providing substantial computer resources.

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

Table 1 Model setup

| Model | CMAQ v4.7.1 | EMEP/MSC-W rv. 4.4 | LOTOS-EUROS v1.8 | SILAM v5.3 |
|---|---|---|---|---|
| **Horizontal resolution** | 18 km | $0.2˚ × 0.2˚$ | $0.3˚ × 0.2˚$ | $0.3˚ × 0.2˚$ |
| **Vertical resolution** | 34 layers up to ~20 km; lowest layer ~20m | 20 layers up to 100 hPa; lowest layer ~ 90m; 3m concentrations derived from the lowest layer values | 3 layers up to 3.5 km; lowest the mixing layer; 25m surface layer for tracking surface concentrations | 8 layers up to ~8km; lowest layer 20 m |
| **Meteo driver** | WRF v3.2.1 | ECMWF | ECMWF | ECMWF |
| **Chemistry scheme** | CB05 | EMEP EmChem09 (Simpson et al., 2012) | TNO CBM-IV | DMAT (Sofiev, 2000) |
| **Aerosol scheme** | aero5 | MARS and VBS (Bergström et al., 2012) | ISORROPIA2 | Extended DMAT (Sofiev, 2000) |
| **Temporal emission profiles** | (Builtjes et al., 2003) | (Simpson et al., 2012) | (Builtjes et al., 2003) | EuroDelta |
| **Vertical emission profiles** | SMOKE plume rise based on (Briggs, 1971) | (Simpson et al., 2012) | EURODELTA (Cuvelier et al., 2007) | (Bieser et al., 2011) |
| **Sea salt emission** | (Spicer et al., 1998). | (Tsyro et al., 2011) | Mårtensson et al., 2003; Monahan et al., 1986 | (Sofiev et al., 2011) |
| **Reference** | (Foley et al., 2010) | (Simpson et al., 2012) | (Schaap et al., 2008, Wichink Kruit et al., 2012) | (Sofiev et al., 2015) |

Table 2. The chemical components of PM computed by the different models, particle sizes, speciation and lumping used in the model simulations. The minus signs indicate that the chemical component was excluded from the computations.

| Component | CMAQ | EMEP | LOTOS-EUROS | SILAM |
|---|---|---|---|---|
| $SO_4^{2-}$ | Aitken, accumulation and coarse modes | $PM_{2.5}$ | $PM_{2.5}$, $PM_{2.5-10}$ | $PM_{2.5}$ |
| $NO_3^-$ | Aitken, accumulation and coarse | $PM_{2.5}$, $PM_{2.5-10}$ | $PM_{2.5}$, $PM_{2.5-10}$ | $PM_{2.5}$, $PM_{2.5-10}$ |
| $NH_4^+$ | Aitken, accumulation and coarse | $PM_{2.5}$ | $PM_{2.5}$ | $PM_{2.5}$ |
| EC | Aitken, accumulation | $PM_{2.5}$ | $PM_{2.5}$ | $PM_{2.5}$ |
| POA (primary organic aerosol) | Aitken, accumulation as total organic mass | $PM_{2.5}$ both carbon and total OA masses | Anthropogenic primary OC included in other primary PM | Anthropogenic primary OC included in other primary PM |
| SOA (secondary organic aerosol) | Accumulation mode As total OA mass | $PM_{2.5}$, both carbon and total OA masses | - | - |
| Sea salt | Accumulation, coarse chemical components computed separately | $PM_{2.5}$, $PM_{2.5-10}$, unspeciated | $PM_{2.5}$, $PM_{2.5-10}$, as chemical components computed separately | Five bins up to 30 µm size, unspeciated |
| Mineral dust (from boundary conditions) | Lumped with unspeciated primary PM | $PM_{2.5}$, $PM_{2.5-10}$ | - | $PM_{2.5}$, $PM_{2.5-10}$ |
| Fire originated aerosol | Fine fraction lumped with primary OC, coarse with unspeciated primary PM | $PM_{2.5}$, $PM_{2.5-10}$ Unspeciated, provided but not included in total PM field | $PM_{2.5}$, $PM_{2.5-10}$ Unspeciated, provided but not included in total PM field | $PM_{2.5}$, $PM_{2.5-10}$ Unspeciated, |
| Benzo[a]pyrene | - | $PM_{2.5}$ | $PM_{2.5}$ | $PM_{2.5}$ |

Table 3. The availability of concentration data for the relevant chemical species from the EMEP network in 2005. TPM – total PM without size limitations.

| Species | $PM_{2.5}$ | $PM_{10}$ | $Na^+$ | $Ca^{2+}$ | $NH_4^+$ | $NH_4^+ +$ $NH_3$ | $NO_3^-$ | $NO_3^- +$ $HNO_3$ | $SO_4^{2-}$ | $SO_2$ | EC/ OC | BaP |
|---|---|---|---|---|---|---|---|---|---|---|---|---|
| Particle size | $PM_{2.5}$ | $PM_{10}$ | TPM | TPM | TPM | Gas + TPM | TPM | Gas + TPM | TPM | Gas | $PM_{2.5}$ and $PM_{10}$ | Gas + TPM |
| Number of stations | 25 | 35 | 26 | 21 | 34 | 45 | 42 | 45 | 73 | 58 | 4 | 8 |

Table 4. The chemical components of PM available from the four EMEP stations that included the EC/OC measurements.

| Station | Temporal resolution | Observed species |
|---|---|---|
| Melpitz (DE0044R, 51.53N, 12.93E) | Daily | $PM_{2.5}$, $PM_{10}$;<br>EC, OC, $NH_4^+$, $NO_3^-$, $SO_4^{2-}$, $Na^+$, Cl, $Ca^{2+}$, Mg, K in $PM_{2.5}$ and $PM_{10}$ |
| Montseny (ES1778R, 41.77N, 2.35E) | One day per week | $PM_{2.5}$;<br>EC, OC, $NH_4^+$, $NO_3^-$, $SO_4^{2-}$, $Na^+$, Cl, $Ca^{2+}$, Mg, K, Si, $CO_3$, Fe, Al in $PM_{2.5}$ and $PM_{10}$ |
| Ispra (IT0004R, 45.8N, 8.63E) | Daily | $PM_{2.5}$, $PM_{10}$;<br>EC, OC, $NH_4^+$, $NO_3^-$, $SO_4^{2-}$ in $PM_{2.5}$ and $PM_{10}$ (no EC observations until 01.05.2005) |
| Birkenes (NO0001R, 58.38N, 8.25E) | Weekly | EC, OC in $PM_{2.5}$ and $PM_{10}$ |
| | Daily | $PM_{2.5}$, $PM_{10}$; $NH_4^+$, $NO_3^-$, $SO_4^{2-}$, $Na^+$, Cl, $Ca^{2+}$, Mg, K in aerosol, no size segregation. |

Table 5. Model-measurement statistics for dry PM for the four models and the two ensemble median models. The colour scale emphasizes the range of values.

| | PM$_{2.5}$ winter (djf) | | obs ave 12.08 | | PM$_{2.5}$ summer (jja) | | obs ave 10.78 | |
|---|---|---|---|---|---|---|---|---|
| | Bias | tCor | sCor | Fac2 | Bias | tCor | sCor | Fac2 |
| CMAQ | -6.41 | 0.48 | 0.53 | 0.49 | -6.15 | 0.54 | 0.68 | 0.30 |
| EMEP | -4.48 | 0.68 | 0.79 | 0.67 | -3.41 | 0.62 | 0.70 | 0.69 |
| LOTOS-EUROS | -3.70 | 0.61 | 0.62 | 0.61 | -6.36 | 0.37 | 0.26 | 0.30 |
| SILAM | -2.10 | 0.65 | 0.86 | 0.66 | -4.56 | 0.52 | 0.59 | 0.46 |
| median | -4.41 | 0.70 | 0.77 | 0.67 | -5.46 | 0.61 | 0.69 | 0.44 |
| medianComp | -4.42 | 0.70 | 0.76 | 0.65 | -2.96 | 0.59 | 0.58 | 0.54 |

| | PM$_{10}$ winter (djf) | | obs ave 16.15 | | PM$_{10}$ summer (jja) | | obs ave 16.53 | |
|---|---|---|---|---|---|---|---|---|
| | Bias | tCor | sCor | Fac2 | Bias | tCor | sCor | Fac2 |
| CMAQ | -7.79 | 0.36 | 0.30 | 0.56 | -9.99 | 0.43 | 0.64 | 0.28 |
| EMEP | -4.53 | 0.55 | 0.56 | 0.66 | -5.55 | 0.59 | 0.77 | 0.7 |
| LOTOS-EUROS | -5.47 | 0.48 | 0.54 | 0.6 | -9.78 | 0.25 | 0.06 | 0.38 |
| SILAM | -4.56 | 0.56 | 0.78 | 0.57 | -7.24 | 0.43 | 0.76 | 0.44 |
| median | -5.82 | 0.60 | 0.62 | 0.65 | -8.69 | 0.51 | 0.70 | 0.45 |
| medianComp | -5.86 | 0.59 | 0.64 | 0.64 | -6.02 | 0.53 | 0.77 | 0.54 |

Notations:

Obs ave – average observed value, mean over all stations, μg/m$^3$.

Bias – absolute bias of the predicted concentrations, mean over all stations (model-measurement, non-scaled, in μg/m$^3$)

tCor – mean temporal correlation of the daily timeseries, mean over all stations

sCor – spatial correlation of the seasonal mean values for the stations

Fac2 – fraction of daily modelled values within a factor of two from the observations

medianComp – sum of the ensemble median fields of the aerosol components

**Table 6 Annual statistics for the PM$_{2.5}$ and PM$_{10}$, dry mass, aerosol bound water added assuming stable state (lower curve of the hysteresis loop) and metastable state (higher curve of the hysteresis loop). ScaledBias - bias divided with the mean observed value, tCor - temporal correlation of the daily values, Fac2 – the fraction of daily values within factor of two from the observed ones. The shading emphasizes the range of the values**

| Species | Model | Dry | | | 50% relative humidity, 20° C, stable | | | 50% relative humidity, 20° C, metastable | | |
|---|---|---|---|---|---|---|---|---|---|---|
| | | Scaled bias | tCor | Fac2 | Scaled bias | tCor | Fac2 | Scaled bias | tCor | Fac2 |
| **PM$_{2.5}$** Ave obs: 11.78 µg/m$^3$ | CMAQ | -0.47 | 0.50 | 0.47 | -0.44 | 0.50 | 0.50 | -0.34 | 0.49 | 0.58 |
| | EMEP | -0.33 | 0.62 | 0.69 | -0.30 | 0.62 | 0.71 | -0.17 | 0.62 | 0.77 |
| | LOTOS-EUROS | -0.40 | 0.46 | 0.51 | -0.34 | 0.43 | 0.54 | -0.26 | 0.45 | 0.58 |
| | SILAM | -0.26 | 0.59 | 0.58 | -0.18 | 0.57 | 0.61 | -0.08 | 0.57 | 0.64 |
| | median | -0.38 | 0.63 | 0.61 | -0.35 | 0.63 | 0.63 | -0.26 | 0.63 | 0.70 |
| | medianComp | -0.30 | 0.60 | 0.62 | -0.28 | 0.60 | 0.64 | -0.17 | 0.60 | 0.71 |
| **PM$_{10}$** Ave obs: 17.09 µg/m$^3$ | CMAQ | -0.49 | 0.46 | 0.49 | -0.40 | 0.42 | 0.53 | -0.29 | 0.41 | 0.59 |
| | EMEP | -0.31 | 0.57 | 0.69 | -0.21 | 0.51 | 0.70 | -0.09 | 0.51 | 0.72 |
| | LOTOS-EUROS | -0.44 | 0.40 | 0.53 | -0.32 | 0.29 | 0.57 | -0.25 | 0.32 | 0.61 |
| | SILAM | -0.34 | 0.54 | 0.54 | -0.24 | 0.50 | 0.58 | -0.16 | 0.51 | 0.60 |
| | median | -0.41 | 0.59 | 0.59 | -0.33 | 0.53 | 0.63 | -0.23 | 0.54 | 0.68 |
| | medianComp | -0.35 | 0.57 | 0.63 | -0.26 | 0.53 | 0.66 | -0.17 | 0.54 | 0.70 |

**Table 7 Annual statistics for the PM components: ScaledBias - bias divided with the mean observed value, tCor - temporal correlation of the daily values, Fac2 – the fraction of daily values within factor of two from the observed ones. The shading emphasizes the range of the values.**

| Species | Model | Scaled bias | tCor | Fac2 | Species | Model | Scaled bias | tCor | Fac2 |
|---|---|---|---|---|---|---|---|---|---|
| **NH$_4$** | CMAQ | -0.08 | 0.55 | 0.49 | **NO$_3$** | CMAQ | -0.12 | 0.35 | 0.47 |
| Ave obs: | EMEP | -0.08 | 0.58 | 0.51 | Ave obs: | EMEP | 0.13 | 0.46 | 0.45 |
| 0.86 µg N/m$^3$ | LOTOS-EUROS | -0.06 | 0.56 | 0.47 | 0.52 µg N/m$^3$ | LOTOS-EUROS | 0.06 | 0.44 | 0.42 |
| | SILAM | -0.16 | 0.55 | 0.37 | | SILAM | 0.06 | 0.44 | 0.39 |
| | median | -0.13 | 0.61 | 0.5 | | median | 0.00 | 0.49 | 0.49 |
| **NH$_3$+NH$_4$** | CMAQ | 0.00 | 0.38 | 0.44 | **NO$_3$+HNO$_3$** | CMAQ | 0.14 | 0.49 | 0.67 |
| Ave obs: | EMEP | -0.06 | 0.45 | 0.59 | Ave obs: | EMEP | 0.24 | 0.49 | 0.56 |
| 1.54 µg N/m$^3$ | LOTOS-EUROS | 0.12 | 0.39 | 0.59 | 0.58 µg N/m$^3$ | LOTOS-EUROS | 0.12 | 0.47 | 0.6 |
| | SILAM | 0.10 | 0.44 | 0.54 | | SILAM | 0.02 | 0.48 | 0.49 |
| | median | 0.01 | 0.47 | 0.6 | | median | 0.10 | 0.54 | 0.65 |
| **NH$_3$** | CMAQ | 0.04 | 0.18 | 0.25 | **HNO$_3$** | CMAQ | 0.21 | 0.34 | 0.43 |
| Ave obs: | EMEP | -0.07 | 0.30 | 0.36 | Ave obs: | EMEP | -0.11 | 0.38 | 0.39 |
| 0.75 µg N/m$^3$ | LOTOS-EUROS | 0.19 | 0.22 | 0.38 | 0.19 µg N/m$^3$ | LOTOS-EUROS | 0.00 | 0.38 | 0.40 |
| | SILAM | 0.32 | 0.30 | 0.40 | | SILAM | -0.53 | 0.32 | 0.32 |
| | median | 0.05 | 0.31 | 0.39 | | median | -0.16 | 0.41 | 0.44 |
| **SO$_4$** | CMAQ | -0.10 | 0.59 | 0.73 | **SO$_2$** | CMAQ | 0.25 | 0.53 | 0.49 |
| Ave obs: | EMEP | -0.18 | 0.58 | 0.57 | Ave obs: | EMEP | 0.23 | 0.47 | 0.48 |
| 0.77 µg S /m$^3$ | LOTOS-EUROS | -0.38 | 0.56 | 0.45 | 0.79 µg S/m$^3$ | LOTOS-EUROS | 0.05 | 0.49 | 0.54 |
| | SILAM | -0.04 | 0.51 | 0.52 | | SILAM | -0.13 | 0.48 | 0.5 |
| | median | -0.23 | 0.63 | 0.63 | | median | 0.04 | 0.55 | 0.54 |
| **Sea salt** | CMAQ | 0.40 | 0.48 | 0.46 | **Mineral dust** | EMEP | -0.75 | 0.29 | 0.29 |
| Ave obs: | EMEP | 0.38 | 0.54 | 0.49 | Ave obs: | SILAM | -0.58 | 0.31 | 0.33 |
| 0.78 µg Na/m$^3$ | LOTOS-EUROS | -0.03 | 0.38 | 0.49 | 0.12 µg Ca/m$^3$ | median | -0.67 | 0.32 | 0.31 |
| | SILAM | 0.08 | 0.44 | 0.48 | | | | | |
| | median | 0.13 | 0.55 | 0.58 | | | | | |
| **EC in PM$_{2.5}$** | CMAQ | -0.61 | 0.51 | 0.35 | **EC in PM$_{10}$** | CMAQ | -0.69 | 0.42 | 0.32 |
| Ave obs: | EMEP | -0.56 | 0.53 | 0.4 | Ave obs: | EMEP | -0.66 | 0.46 | 0.35 |
| 1.08 µg C/m$^3$ | LOTOS-EUROS | -0.34 | 0.51 | 0.44 | 1.32 µg C/m$^3$ | LOTOS-EUROS | -0.48 | 0.39 | 0.44 |
| | SILAM | -0.17 | 0.61 | 0.4 | | SILAM | -0.35 | 0.45 | 0.38 |
| | median | -0.45 | 0.6 | 0.38 | | median | -0.58 | 0.49 | 0.37 |
| **OC in PM$_{2.5}$** | CMAQ | -0.80 | 0.52 | 0.26 | **OC in PM$_{10}$** | CMAQ | -0.85 | 0.36 | 0.18 |
| Ave obs: | EMEP | -0.25 | 0.54 | 0.6 | Ave obs: | EMEP | -0.37 | 0.46 | 0.52 |
| 3.61 µg C/m$^3$ | median | -0.52 | 0.54 | 0.61 | 4.78 µg C/m$^3$ | median | -0.61 | 0.46 | 0.48 |

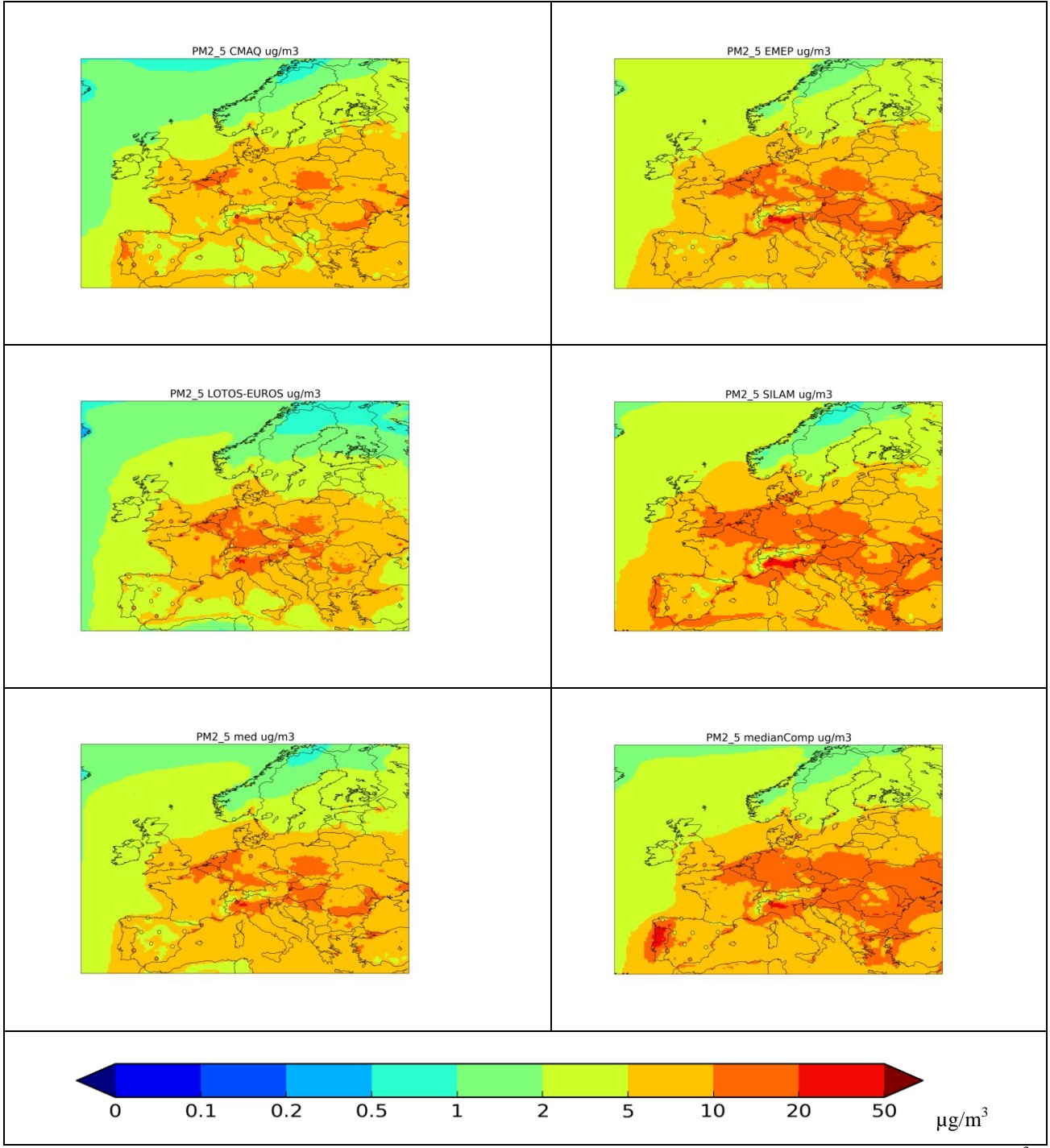

Figure 1. Annual mean dry PM$_{2.5}$ concentration predicted by the models, their median and medianComp [μg PM m$^{-3}$]. The dots show the annual mean observed values in EMEP stations (only the stations with observations available for at least 75% of the time are shown)

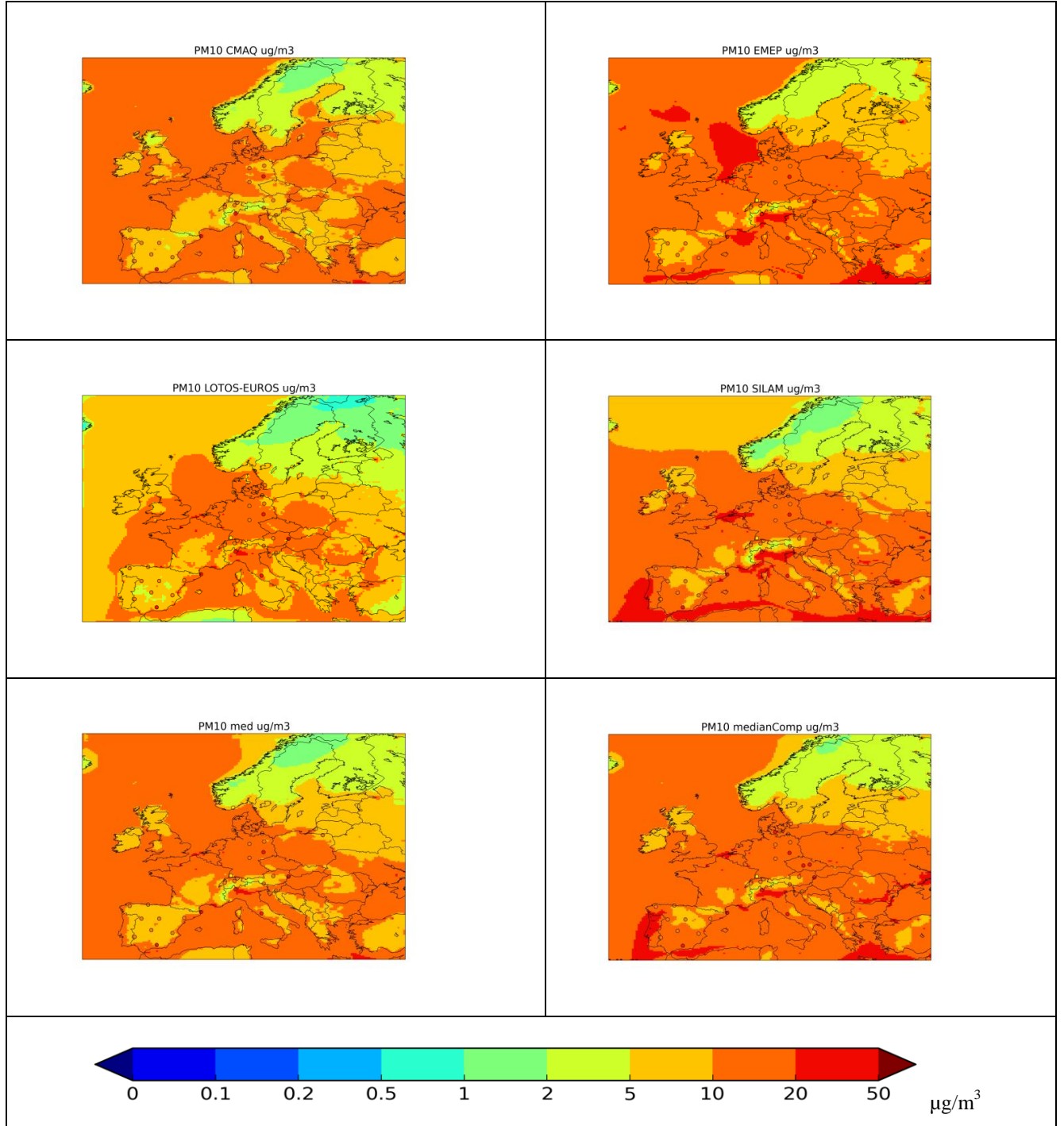

Figure 2. Annual mean dry PM$_{10}$ concentration predicted by the models, their median and medianComp [µg PM m-$^3$]. The dots show the annual mean observed values in EMEP stations (only the stations with observations available for at least 75% of the time are shown).

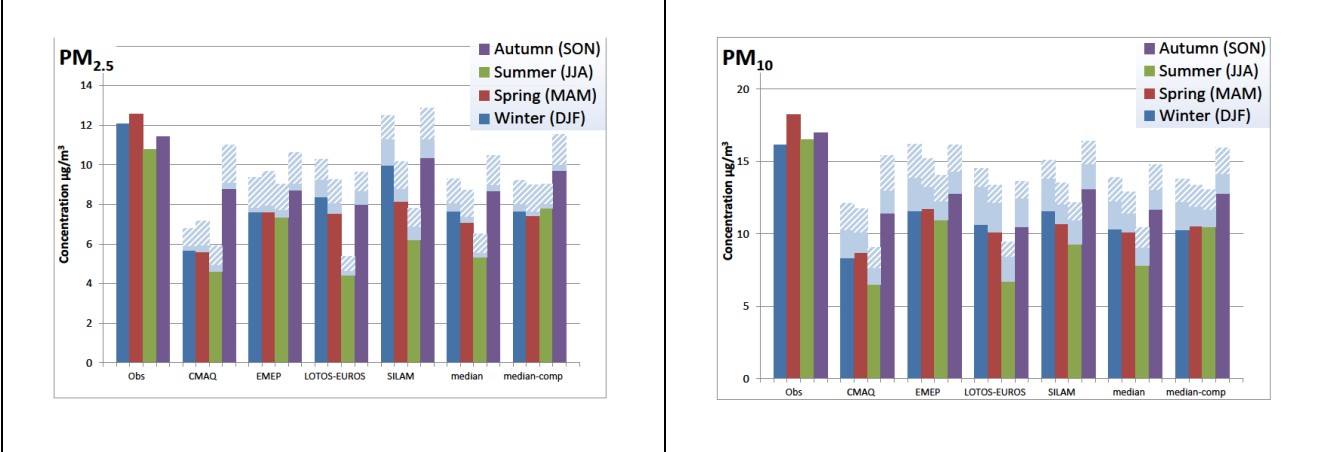

Figure 3. Observed and predicted seasonal concentrations of $PM_{2.5}$ (left) and $PM_{10}$ (right), mean over the EMEP stations [µg PM m$^{-3}$]. The light blue part shows the aerosol-bound water amount at the filter weighting conditions (50% relative humidity, 20°C), estimated with ISORROPIA2 based on the modelled aerosol composition. The solid light blue shows the water content in stable case (the lower curve of the hysteresis loop) and the striped part in metastable case (the upper branch of the hysteresis loop), when the crystallization has not occurred to aerosol coming from more humid conditions.

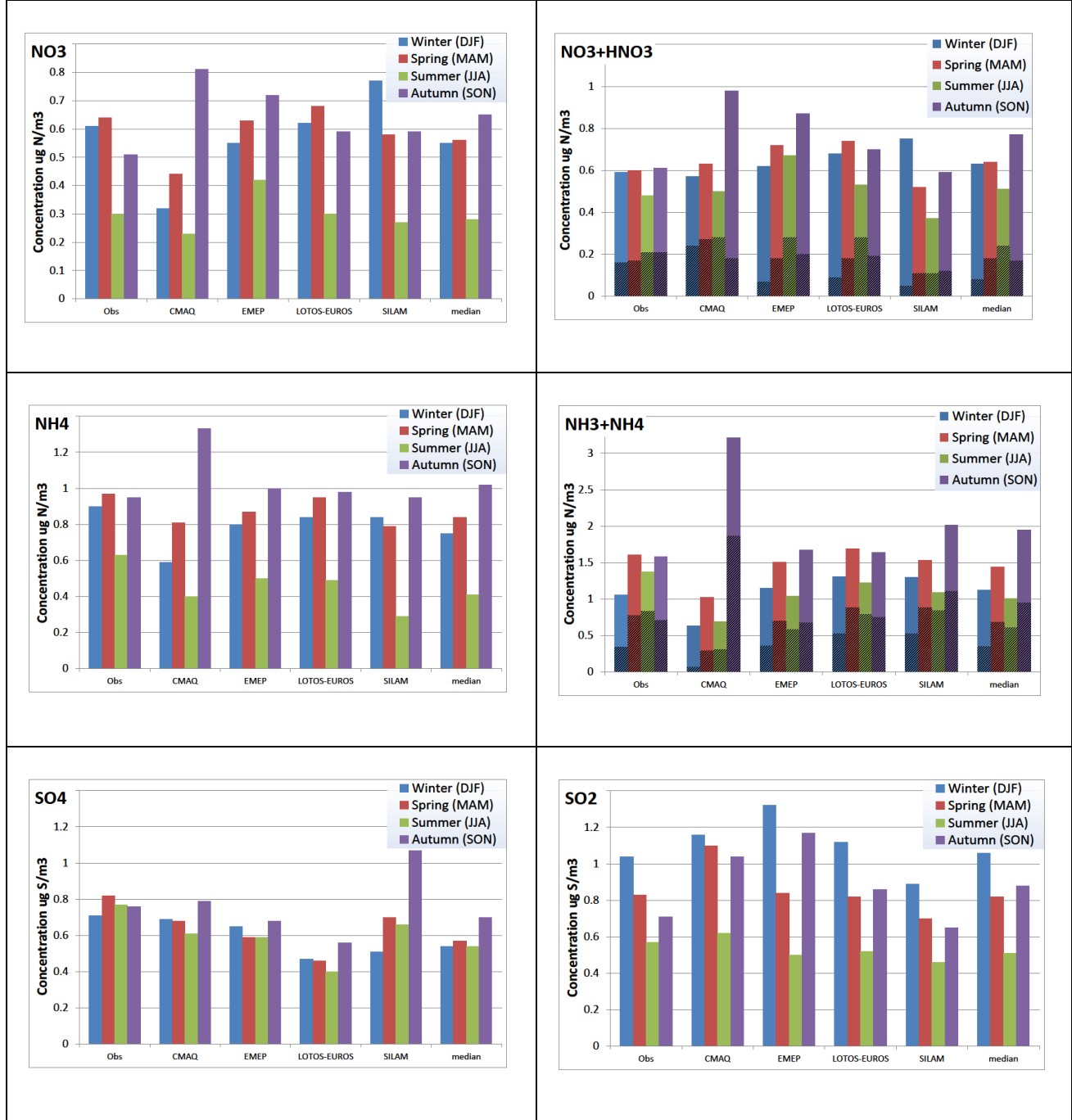

Figure 4. Observed and predicted seasonal concentrations of secondary inorganic aerosols and their precursors, mean over the EMEP stations [µg S/N m$^{-3}$ ]. Shaded part shows the concentration of the gas phase species HNO$_3$ and NH$_3$. Only the stations where at least two of the gas, aerosol and their sum were observed, so that the gas phase fraction could be estimated, are included in the averaging. .

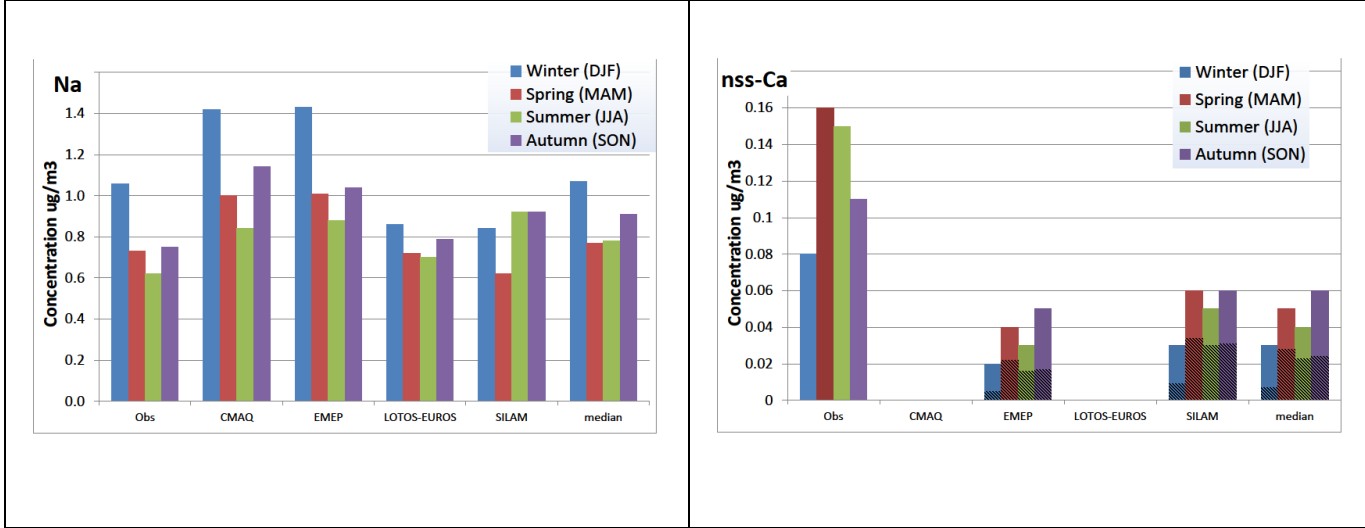

Figure 5. Observed and predicted seasonal concentrations of sodium and non-seasalt calcium in aerosol, mean over the EMEP stations [μg m$^{-3}$]. Modelled Na$^+$ concentrations are based on sea salt containing 30.8% Na$^+$. Model values of non-seasalt calcium assume 10% Ca$^{2+}$ content of desert dust (shaded bottom part of the columns) and 3.5% calcium content of non-carbonaceous primary anthropogenic PM (the non-shaded upper part).

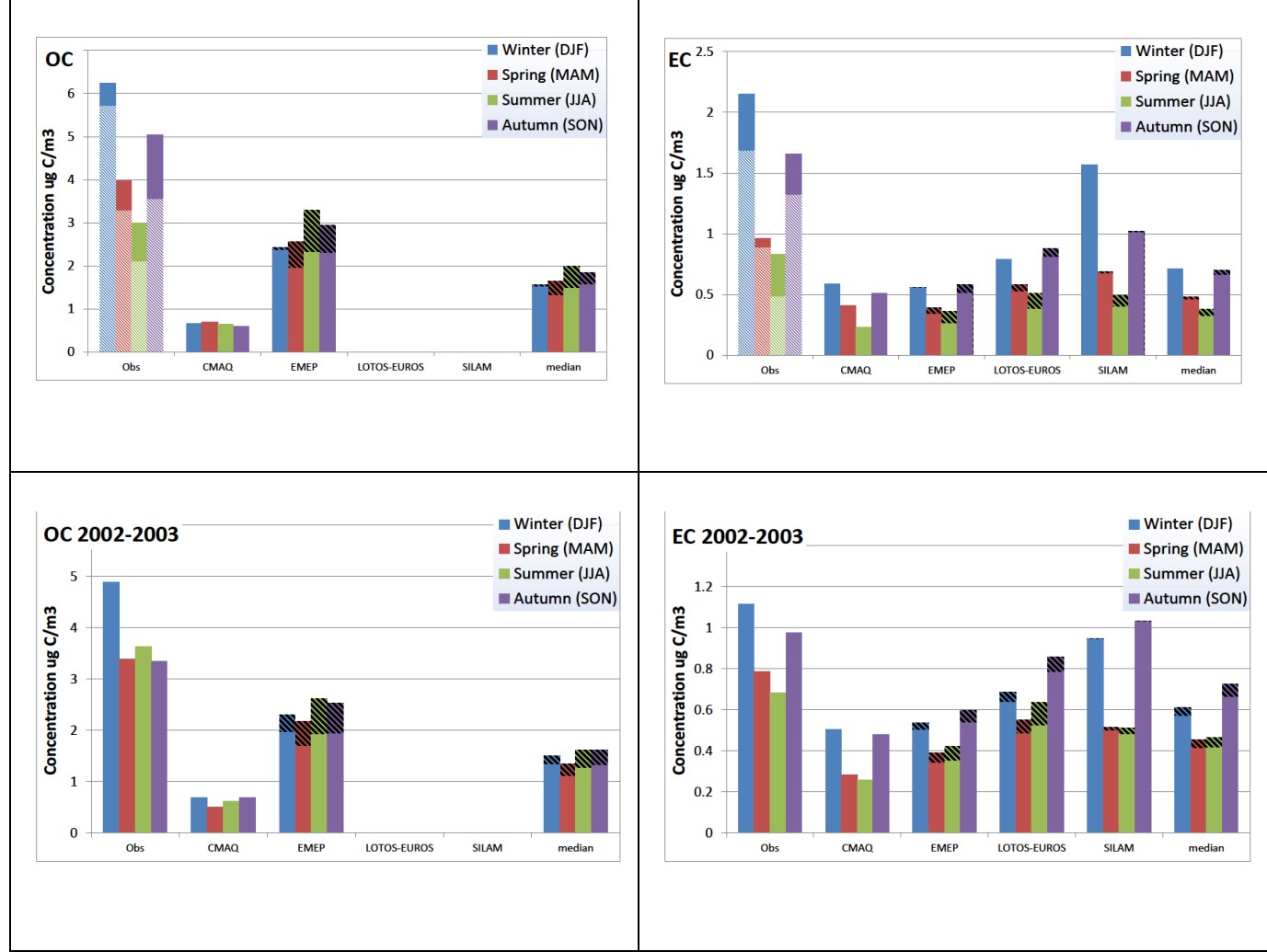

Figure 6. Observed and predicted seasonal concentrations of carbonaceous aerosols, mean over the EMEP stations [µg m-3 ]. The panels on the left-hand and right-hand sides represent OC and EC, respectively. The upper row: 2005, data from 4 stations, for the observations the lighter shading marks the concentration in PM2.5, whole column the concentration in PM10; the lower row: EMEP 2002-2003 campaign, observations of OC and EC in PM10. Dark shaded part shows the contribution from wild land fires (not separated for CMAQ OC, missing for CMAQ EC).

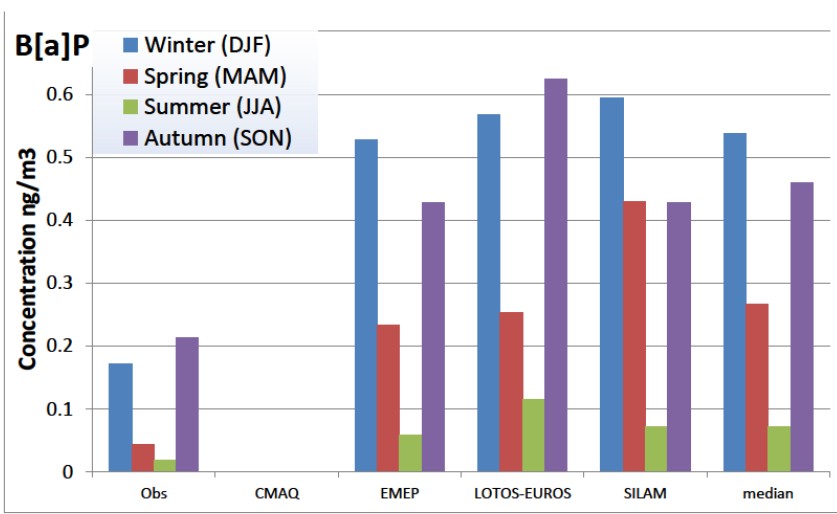

**Figure 7 Observed and predicted seasonal concentrations of benso(a)pyrene, mean over the EMEP stations in 2005 [ng/m³]**

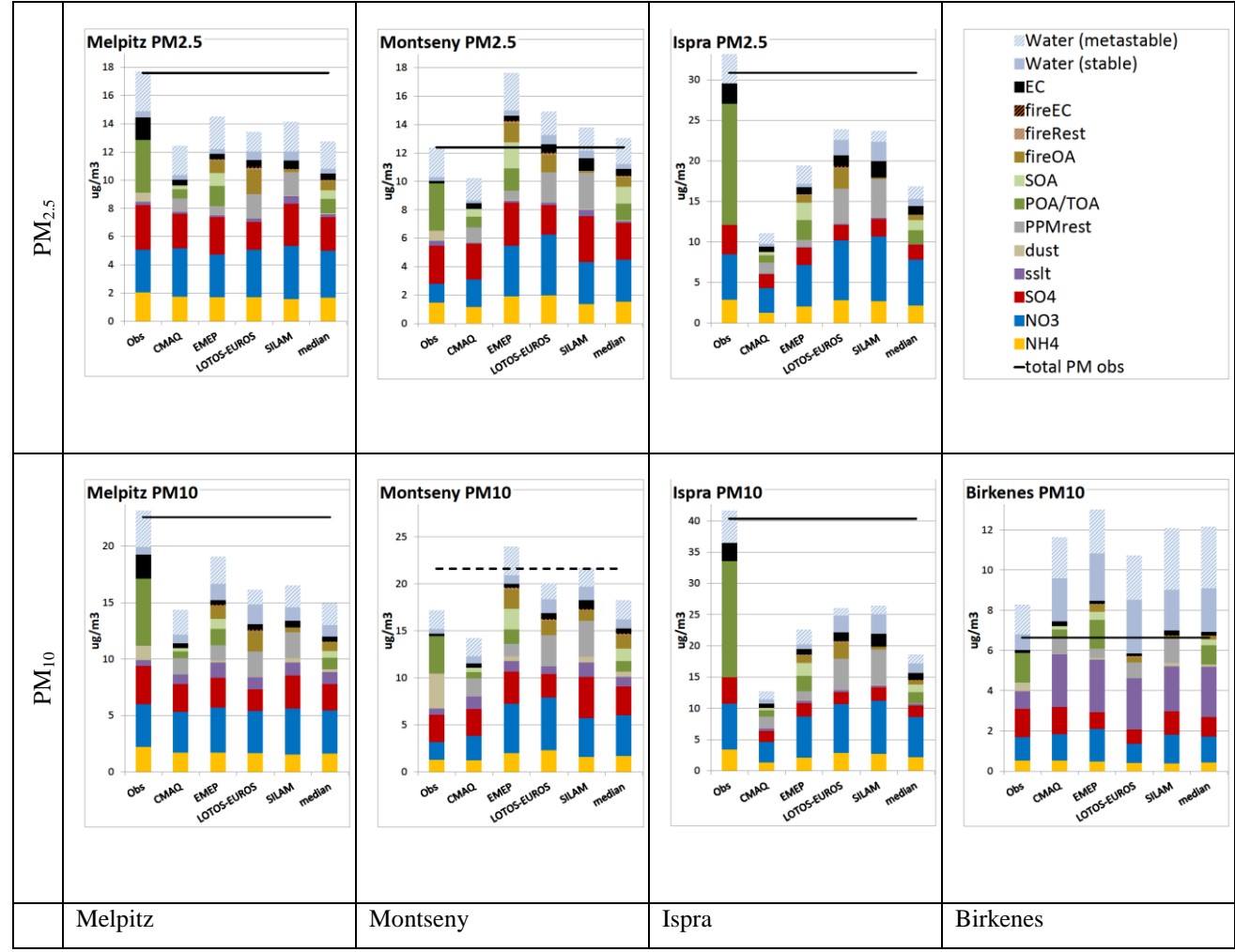

Figure 8. Aerosol chemical composition measured and modelled at four stations. Upper row – PM$_{2.5}$, lower row – PM$_{10}$

Water – aerosol water content at 50%RH and 20°C, computed with ISORROPIA2 based on observed or modelled aerosol composition.
metastable – particle is assumed to be in supersaturated liquid state if the relative humidity is below its deliquescence point
stable – particle is assumed solid if the relative humidity is below its deliquescence point
EC – elemental carbon from anthropogenic emissions
fireEC - elemental carbon from wild-land fire emissions, 5% of fire emitted PM
fireRest – mineral PM from wild-land fire emissions, 5% of fire emitted PM
fireOA– organic aerosol from wild-land fire emissions, 90% of fire emitted PM
SOA – secondary organic aerosol
POA/TOA – the primary part of organic aerosol for the models, total organic aerosol for the observations (OC * 1.6)
PPMrest – the unspeciated part of the modelled primary anthropogenic PM
Dust – modelled desert dust, observed non-sea-salt Ca$^{2+}$ x 10
Sslt – sea-salt, observed Na$^+$ + Cl$^-$
SO4, NO3, NH4 – secondary inorganic aerosols
Total PM obs – observed total PM$_{2.5}$ and PM$_{10}$.
* PM$_{10}$ observations were not available for Montseny station. The dotted line marks an estimate calculated by averaging PM$_{10}$ observations from the nearest
EMEP stations (ES0010R, ES0014R).
* Na observations were not available in Ispra and were excluded from ISORROPIA input.