# Peer review of "Evaluation of the performance of four chemical transport models in predicting the aerosol chemical composition in Europe in 2005"

_Atmospheric Chemistry and Physics, 2015_

## Referee Comment (RC1) · Anonymous Referee #2 · 21 Feb 2016

The manuscript by Prank et al. presents a new intercomparison among four regional models at the European scale for year 2005. The intercomparison focuses on particulate matter and results are presented in comparison to available measurements of the PM10-PM2.5 mass and several PM chemical components carried out at 12 EMEP stations. In this reviewer opinion, the manuscript contains a useful update on the operational validation of the models involved in the study, and indicates major uncertainties and main directions for further models improvement.

The presentation of result is clear and the manuscript generally well written, but a little effort in order to make it more concise is recommended. Before publication, I encourage the authors implementing the following comments/suggestions:

[Figure]

- please double check the reference list: I checked the first three citations, and they are all missing in the list at the end of the manuscript.

- I find too much overlap and redundancy in the introductory part on model-to-obs uncertainties (page 2-4) and the discussion (section 4). My impression is that a quite long list of possible reasons for model-to-obs disagreement is presented in both parts, but never really demonstrate them for the specific simulations presented here. I thus suggest to shorten both the introduction and the discussion, and possibly move all the model-to-obs issue directly into the discussion section.

- page 4, lines 7-10: I do not completely agree with this final statement. From my understanding, the "main reasons behind model-measurements" are not clearly identified in the study. I would better state that the model error regarding the PM simulation is characterized against available measurments.

- Figure S4: please define PPMr in the caption.

- page 8, lines 9-13: there seems to be some inconsistency between the major components illustrated in figures S5-S8 and values given in Table 6. In particular, from Table 6 one would say that carbonaceous aerosol are the major fraction of PM, not secondary organic aerosol. Please clarify.

- page 8, line 31: could a poor correlation coefficient for NO3- be related to a pulsed behaviour of the aerosol nitrate production in the PBL, as recently described in this paper:

Curci, G., Ferrero, L., Tuccella, P., Barnaba, F., Angelini, F., Bolzacchini, E., Carbone, C., Denier van der Gon, H. A. C., Facchini, M. C., Gobbi, G. P., Kuenen, J. P. P., Landi, T. C., Perrino, C., Perrone, M. G., Sangiorgi, G., and Stocchi, P.: How much is particulate matter near the ground influenced by upper-level processes within and above the PBL? A summertime case study in Milan (Italy) evidences the distinctive role of nitrate, Atmos. Chem. Phys., 15, 2629-2649, doi:10.5194/acp-15-2629-2015, 2015.

- page 9, lines 7-8: the discussion on HNO3 bias is made difficult by the fact that HNO3 is not shown alone in the figure. Could this be shown, or the results commented on the NO3+HNO3 concentration that the reader may actually directly see in the plots?

- page 9, line 10: it is not completely true that the seasonal cycle is not reproduced by all SIA, e.g. NO3 and NH4 are reproduced quite well.

- page 9, line 16: "... overestimate the temperature dependence ..." suggest to rephrase with "... have an exaggerated temperature dependece ..." to avoid confusion with over-estimated/underestimated resulting PM values.

- page 9: in general, natural PM seems to be a major factor contributing to the spring PM maximum: may you confirm that (or not)?

- page 10, lines 16-17: perhaps could be useful the discussion on EC lifetimes presented in this paper:

Wang, X., Heald, C. L., Ridley, D. A., Schwarz, J. P., Spackman, J. R., Perring, A. E., Coe, H., Liu, D., and Clarke, A. D.: Exploiting simultaneous observational constraints on mass and absorption to estimate the global direct radiative forcing of black carbon and brown carbon, Atmos. Chem. Phys., 14, 10989-11010, doi:10.5194/acp-14-10989-2014, 2014.

---

## Referee Comment (RC2) · Anonymous Referee #1 · 12 Mar 2016

The study addresses an interesting issue relevant with the performance of European scale models in the representation of the chemical composition of PM in the near surface air. The scientific tools used to address the issue are well documented and methodology is scientific sound. Clarifications are necessary to describe better the methodological steps. Also, the paper provides some answers on the reasons for the better or worse model performance; however most of them have been discussed in previous publications. Also the uncertainties in the model evaluation presented in the paper are only theoretically addressed but not quantified. Interesting is the issue of the water mass included in the aerosols, which in most cases is not accounted for in the validation of PM model results; this issue is also theoretically discussed in the

manuscript.

Following are some suggestion and comments for the submitted manuscript:

1) Section 2.3: The authors should present a table in which all the emitted and simulated PM species will be presented in detail for each model. The existing Table 2 cannot explain how sea salt is speciated in model simulations (is it simulated explicitly as Na and Cl?). Similar is the comment for fire PM species. Usually, fire PM emissions are simulated as OC and EC emissions and I would expect these emissions to have a contribution to OC and EC levels. How is this issue addressed in the different models? If fire PM emissions were speciated as OC and EC then how was it possible to distinguish between anthropogenic OC/EC and fire-related OC/EC? The models simulate OC or OA? Also in Table 2, the PM size is not presented (PM10 versus PM2.5).

2) Section 2.4: Please present in Table 3 the size of the PM measured (PM10 versus PM2.5).

3) Section 2.5: The method for the estimation of nss-Ca levels from the model results is not clear. Are the simulated nss-Ca concentrations estimated as the sum of the 10% of dust concentrations plus 3.5% of the unspeciated other primary PM concentrations? The authors should also make clear in the whole manuscript (e.g. in Table 6) that it is not the mineral dust model results that are evaluated but the nss-Ca values (including both the desert dust and the anthropogenic contribution).

4) Section 3.1, page 9, lines 7-18: The authors explained the bad model performance in Schauinsland Mountain station as the result of model spatial resolution and high station altitude. The position and altitude of the stations should be provided in the supplement. Are there other stations of similar altitude in which the models performance is not as bad in the Schauinsland Mountain station?

5) Section 3.1, page 10, lines 2-4: Explain the reasons for models' different ability in simulating PM2.5 and PM10 in summer and winter.

6) Section 3.1, page 10, line 11: The authors state: "The medianComp fully includes SOA, desert dust and fire-induced PM." This is not true since according to Table 2, SOA are not included in LOTOS-EUROS and in SILAM. Also BCs dust is not included in LOTOS-EUROS.

7) Section 3.1, page 10, lines 16-18: The authors explain that the differences between median and medianComp are due to desert dust. However, according to Table 2, it is only the fire originated PM that were not included in total PM while BCs dust was included all CMAQ, EMEP and SILAM.

8) Section 3.2, Table 6: Which are the size bins of the PM species validated (comment mostly for sea salt and mineral dust). The words "mineral dust" should be replaces with "nss-Ca". Why evaluation is presented explicitly only for SO2 and not for other PM precursor gas compounds like NH3 or NOx?

9) Section 3.2.3: The performance of CMAQ, can it be explained by the fact that fire emissions are included in other PM?

10) Section 3.2.5: Any comments on OA comparison with observations should be based on CMAQ and EMEP results since the other models include OC in other PM. Similar remark for dust straightforwardly included only in EMEP and SILAM.

11) Section 3.2.5., page 15, line 7: Correct OA with OC.

12) Section 3.2.5., page 15, line 11: Please add that also the OC is for some models included in primary aerosols (as the case for LOTOS-EUROS and SILAM)

13) Section 3.2.5., Figure 8: a) suggestion: add the simulated SOA and POA and compare them with the observed TOA, b) why fire PM are presented separately and not speciated in OC and EC so as to allow better comparison with observations? (see also comment 1). In the legend of Figure 8 the PM species are not presented in the same order as they appear in the plots.

14) Section 4.1: This section provides a theoretical description of the possible reasons for overestimations or underestimations in the model results. However, there is no quantification of the uncertainties. For example, how much is the uncertainty introduced in the model validation because of the assumption that dust and anthropogenic mineral aerosols have a 10% and 3.5 Ca content respectively? Clarifications also are necessary in the last paragraph of this section in which the authors discuss about the absence of fire emissions from the computations while in Table 2 fire originated PM are presented as included in the simulations of all models.

15) Section 4.3: It is a very interesting section. The authors should present results on the models' improved performance when water is accounted for (even in the four stations providing a complete set of PM measurements; after all section 3.2.5 was based on the measurements in these few stations).

16) Figure 6: Correct in the lower-right plot the OC to EC.

17) Figure S6: Was CMAQ excluded from the average since dust in CMAQ was included in other PM?

18) Figure S7: Was LOTOS-EUROS and SILAM excluded from OA mean values since OA in these modes were included in other PM? The Figure is not commented in the text of the manuscript.

19) Figure S8: Please check comment 1. How fire emissions were simulated and consequently presented in the Figure for PM? If speciated to OC and EC emissions, how fire-related OC and EC concentrations were distinguished from anthropogenic OC and EC concentrations?

---

## Author Comment (AC1) · 14 Mar 2016

**Answer to the comments of Anonymous Referee #2**

We would like to thank the reviewer for their comments (marked blue in the following text). Here are our answers:

*- please double check the reference list: I checked the first three citations, and they are all missing in the list at the end of the manuscript.*

Thanks to the reviewer for noticing this, all references from the first few paragraphs had indeed disappeared from the reference list. The reference list has been fixed.

*- I find too much overlap and redundancy in the introductory part on model-to-obs uncertainties (page 2-4) and the discussion (section 4). My impression is that a quite long list of possible reasons for model-to-obs disagreement is presented in both parts, but never really demonstrate them for the specific simulations presented here. I thus suggest to shorten both the introduction and the discussion, and possibly move all the model-to-obs issue directly into the discussion section.*

The model-observation comparison issues will be moved to discussion.

*- page 4, lines 7-10: I do not completely agree with this final statement. From my understanding, the "main reasons behind model-measurements" are not clearly identified in the study. I would better state that the model error regarding the PM simulation is characterized against available measurements.*

The sentence will be restated according to the reviewer's suggestion.

*- Figure S4: please define PPMr in the caption.*

Done

*- page 8, lines 9-13: there seems to be some inconsistency between the major components illustrated in figures S5-S8 and values given in Table 6. In particular, from Table 6 one would say that carbonaceous aerosol are the major fraction of PM, not secondary organic aerosol. Please clarify.*

There are several reasons for the noticed discrepancies. Firstly, the concentrations are given in different units. Table 6 reports the annual mean concentrations observed by the EMEP network, averaged over all the stations. They are given in the units the observations are provided – micrograms of nitrogen, sulphur, sodium, calcium or carbon. The model median maps on the figures S5-S8 follow the speciation in the models and are in the total mass of the component ($NO_3$, $NH_4$, $SO_4$, sea salt, mineral dust, organic aerosol, fire-emitted PM). Converting the observed values to the total mass of the components, we get the sum of the secondary inorganic species very close to the carbonaceous ones in $PM_{2.5}$. Taking into account that the models underestimate substantially the carbonaceous part, the model maps probably overestimate the SIA fraction in PM. Additional differences are introduced by sampling the map only at the station locations. For the SIA species the station network is well covering and representative, while the carbonaceous components were measured in only four stations.

Clarifications will be added.

*- page 8, line 31: could a poor correlation coefficient for NO3- be related to a pulsed behaviour of the aerosol nitrate production in the PBL, as recently described in this paper: Curci, G., Ferrero, L., Tuccella, P., Barnaba, F., Angelini, F., Bolzacchini, E., Carbone, C., Denier van der Gon, H. A. C., Facchini, M. C., Gobbi, G. P., Kuenen, J. P. P., Landi, T. C., Perrino, C., Perrone, M. G., Sangiorgi, G., and Stocchi, P.: How much is particulate matter near the ground influenced by upper-level processes within and above the PBL? A summertime case study in Milan (Italy) evidences the distinctive role of nitrate, Atmos. Chem. Phys., 15, 2629-2649, doi:10.5194/acp-15-2629-2015, 2015.*

The paper of Curci et al presents a detailed study on nitrate formation in Po Valley and how its concentration is influenced by the boundary layer processes and temperature and humidity vertical profiles. It is indeed an interesting question, how well these effects are taken into account in the regional models. The models use a range of different algorithms for SIA creation, such as thermodynamic equilibrium model ISORROPIA2 in LOTOS-EUROS or separate equilibrium computations for $NH_4NO_3$ and a parameterization for coarse $NO_3$ production on sea salt particles in SILAM and EMEP. Generally these should be capable of simulating the profile of $NO_3$ creation and brake up, provided that the model vertical resolves the temperature and relative humidity profiles. However, this can easily not be the case - for instance LOTOS-EUROS has a single layer representing the whole boundary layer and SILAM layers at boundary layer top can easily be 500-1000 meters thick.

[Figure]

**Figure 1** Summer time concentration profiles of $NH_4NO_3$ from SILAM model, west-east and south-north cross-sections of Europe, selected to cut through Po Valley.

While the low vertical resolution of the gathered model data does not allow for a detailed analysis of this effect in the whole dataset, elevated plumes of $NH_4NO_3$ are clearly visible in SILAM model output (Figure 1), mainly because the efficient dry deposition of $HNO_3$ depletes its concentration in the near-surface layers making $NH_4NO_3$ break to the gaseous compounds. Thus, the formation of the residual layers and mixing them down in the morning and the ability of the models to reproduce these effects can have an impact on the simulation quality. However, as the EMEP aerosol composition observations have daily resolution, it's

hard to tell, how much these processes influence the model-measurement correlation compared with all the other uncertainties related to $NH_4NO_3$ formation, such as the temperature dependence of $NH_3$ emission.

The discussion of the uncertainties in $NO_3$ modelling will be extended in the manuscript.

*- page 9, lines 7-8: the discussion on HNO3 bias is made difficult by the fact that HNO3 is not shown alone in the figure. Could this be shown, or the results commented on the NO3+HNO3 concentration that the reader may actually directly see in the plots?*

The reason for selecting $HNO_3+NO_3$ as the presented variable was, that there are more stations in EMEP network that measure the sum than there are those that measure $HNO_3$ alone, also the sum is measured with higher accuracy, while the HNO3 concentration is more prone to observation artefacts .

Figure 2 shows $HNO_3+NO_3$ together with the contribution from $HNO_3$ (shaded). On annual average level all models overestimate the sum by 5-35 %. EMEP and LOTOS-EUROS accurately predict the $HNO_3$ fraction in the sum, while SILAM underestimates and CMAQ overestimates it by 10%. However, the models do not reproduce well the seasonal variations in $HNO_3$ concentration.

[Figure]

**Figure 2 Observed and predicted seasonal concentrations of $HNO_3+NO_3$, mean over the EMEP stations [µg N m$^{-3}$]. Shaded part shows the concentration of $HNO_3$. Only the stations where at least two of $HNO_3$, $NO_3$ and $HNO_3+NO_3$ were observed, so that the $HNO_3$ fraction could be estimated, are included in the averaging.**

The figure and related discussion will be updated in the paper.

*- page 9, line 10: it is not completely true that the seasonal cycle is not reproduced by all SIA, e.g. NO3 and NH4 are reproduced quite well.*

Will be restated

*- page 9, line 16: "... overestimate the temperature dependence ..." suggest to rephrase with "... have an exaggerated temperature dependece ..." to avoid confusion with overestimated/underestimated resulting PM values*

Done

*. - page 9: in general, natural PM seems to be a major factor contributing to the spring PM maximum: may you confirm that (or not)?*

Of the natural aerosols considered (Figure 5 in the manuscript), sea salt exhibits no spring peak - elevated $Na^+$ concentrations are observed only in winter and the models reproduce this behaviour. Dust, on the other hand, seems to be contributing – observed nss-Ca concentrations peak in spring and so do the modelled Saharan dust concentrations. Previous studies about Saharan dust have found the emissions peaking at spring (Fiedler et al., 2013; Laurent et al., 2008). Additionally to desert dust there are other possible reasons for elevated crustal aerosol concentration in spring, such as agricultural activities and vehicle caused erosion of roads - in colder regions where the roads are sanded against slipperiness high dust emissions occur when the road conditions get dry in spring.

Considering the biogenic primary particles, pollens are abundant in air in spring - maximum concentrations can reach a few tens of thousands of pollen/m3, equivalent to tens of µg/m3. However, their size is mostly too large to make them relevant for $PM_{10}$ (~20 µm for abundant early spring flowering species like birch). Evidence exist, that pollen grains can break in atmosphere and produce particles in size range relevant for even $PM_{2.5}$, however, such particles are unlikely to be abundant enough to be noticeable in the PM budget. Also the fungal spores become abundant only in summer – high concentrations can be observed from June to October.

Regarding the anthropogenic contribution, all observed secondary inorganic species show spring maxima.

*- page 10, lines 16-17: perhaps could be useful the discussion on EC lifetimes presented in this paper: Wang, X., Heald, C. L., Ridley, D. A., Schwarz, J. P., Spackman, J. R., Perring, A. E., Coe, H., Liu, D., and Clarke, A. D.: Exploiting simultaneous observational constraints on mass and absorption to estimate the global direct radiative forcing of black carbon and brown carbon, Atmos. Chem. Phys., 14, 10989-11010, doi:10.5194/acp- 14-10989-2014, 2014.*

The paper of Wang et al reports shorter EC lifetimes to be more consistent with observations and a similar conclusion was reached by Samset et al., (2014) - both studies found that shorter lifetime was necessary for reproducing the EC vertical profiles and low concentrations in remote regions. This result somewhat contradicts with the current model intercomparison, where SILAM was found to best reproduce the observed EC concentrations, and longer EC lifetime due to slower deposition in that model was assumed as the main reason for the model-to-model differences. Also the temporal correlation with 2005 observations

and spatial correlation between the models 2005 average EC and EC observed during the 2002-2003 EMEP campaign is no worse for SILAM than it is for the other models, so there is no clear indication that the slower deposition would not be consistent with the surface EC observations in European scale. However, as indications were found of strong underestimation of EC emission, the slower deposition in SILAM is likely to be compensating for the missing emissions. Observations of vertical profiles and concentrations in more remote locations would be necessary for investigating this issue; unfortunately such were not available in Europe in 2005.

Discussion of EC lifetimes will be added to the manuscript.

Fiedler, S., Schepanski, K., Heinold, B., Knippertz, P., Tegen, I., 2013. Climatology of nocturnal low-level jets over North Africa and implications for modeling mineral dust emission. J. Geophys. Res. Atmos. 118, 6100–6121. doi:10.1002/jgrd.50394

Laurent, B., Marticorena, B., Bergametti, G., Leon, J.F., Mahowald, N.M., 2008. Modeling mineral dust emissions from the Sahara desert using new surface properties and soil database. J. Geophys. Res. Atmos. 113, 1–20. doi:10.1029/2007JD009484

Samset, B.H., Myhre, G., Herber, A., Kondo, Y., Li, S.M., Moteki, N., Koike, M., Oshima, N., Schwarz, J.P., Balkanski, Y., Bauer, S.E., Bellouin, N., Berntsen, T.K., Bian, H., Chin, M., Diehl, T., Easter, R.C., Ghan, S.J., Iversen, T., Kirkevag, A., Lamarque, J.F., Lin, G., Liu, X., Penner, J.E., Schulz, M., Seland, Skeie, R.B., Stier, P., Takemura, T., Tsigaridis, K., Zhang, K., 2014. Modelled black carbon radiative forcing and atmospheric lifetime in AeroCom Phase II constrained by aircraft observations. Atmos. Chem. Phys. 14, 12465–12477. doi:10.5194/acp-14-12465-2014

---

## Author Response (AR1)

**Response to reviews**

We would like to thank the referees for their detailed and constructive comments (marked blue in the following text). Our answers are in black and the changes to the manuscript are marked in italic.

**Answers to the comments of Anonymous Referee #1**

*1) Section 2.3: The authors should present a table in which all the emitted and simulated PM species will be presented in detail for each model. The existing Table 2 cannot explain how sea salt is speciated in model simulations (is it simulated explicitly as Na and Cl?). Similar is the comment for fire PM species. Usually, fire PM emissions are simulated as OC and EC emissions and I would expect these emissions to have a contribution to OC and EC levels. How is this issue addressed in the different models? If fire PM emissions were speciated as OC and EC then how was it possible to distinguish between anthropogenic OC/EC and fire-related OC/EC? The models simulate OC or OA? Also in Table 2, the PM size is not presented (PM10 versus PM2.5).*

 - LOTOS-EUROS and CMAQ calculate the sea salt components separately, in SILAM and EMEP sea salt is not speciated and is emitted and transported as whole. Standard sea salt composition is assumed when comparing the total sea salt concentration with Na observations with 30.8% sodium content by dry mass.

 - The fire PM originated from IS4FIRES (Sofiev et al., 2009), which provides unspeciated $PM_{10}$ and $PM_{2.5}$ emissions. In SILAM, EMEP and LOTOS-EUROS the emitted PM was transported as a separate field of unspeciated particulates, in CMAQ the fine fraction was included in primary OA and the coarse fraction in coarse primary PM. The fire OA in CMAQ cannot be distinguished from the anthropogenic OA and fire EC was not included in that model. In the other models the fire PM can be further speciated as post-processing following Akagi et al., (2011) or Andreae and Merlet, (2001). On average these papers suggest roughly 5% EC and 50% OC content for fire emitted aerosol, the rest mainly consisting of non-carbon atoms in the organic compounds and some inorganics (up to 5%).

 - CMAQ provided total organic aerosol mass; EMEP model calculates both OC and OA.

*Table 2 in the paper has been changed to include the speciation and sizes of the aerosol components computed by each model. Clarifications have been added to Section 2.5 (Model measurement comparison). Wild-land fire emitted contributions to OC and EC have been added to comparisons and discussion of them has been extended.*

*2) Section 2.4: Please present in Table 3 the size of the PM measured (PM10 versus PM2.5).*

The inorganic species were measured mostly in total aerosol without size limits; concentration in $PM_{10}$ was used where available. In 2005 EC and OC were measured in both $PM_{2.5}$ and $PM_{10}$. The 2002-2003 campaign observed EC and OC in $PM_{10}$.

*Table 3 has been updated to include this information.*

*3) Section 2.5: The method for the estimation of nss-Ca levels from the model results is not clear. Are the simulated nss-Ca concentrations estimated as the sum of the 10% of dust concentrations plus 3.5% of the unspeciated other primary PM concentrations? The authors should also make clear in the whole manuscript (e.g. in Table 6) that it is not the mineral dust model results that are evaluated but the nss-Ca values (including both the desert dust and the anthropogenic contribution).*

The simulated nss-Ca concentrations are indeed estimated as the sum of the 10% of dust concentrations plus 3.5% of the unspeciated other primary PM concentrations. *Clarifications were added.*

*4) Section 3.1, page 9, lines 7-18: The authors explained the bad model performance in Schauinsland Mountain station as the result of model spatial resolution and high station altitude. The position and altitude of the stations should be provided in the supplement. Are there other stations of similar altitude in which the models performance is not as bad in the Schauinsland Mountain station?*

There are other high altitude stations, located in Alps and in Spain. The Schauinsland station was pointed out in the paper, because it was the only station where the models consistently overestimated the PM concentration. Also the temporal correlations between the models and observations are the lowest there for both $PM_{10}$ and $PM_{2.5}$. The models' performance does degrade also in the other high stations - there is a strong negative correlation between the station altitude and the models' temporal correlation coefficients for both $PM_{10}$ and $PM_{2.5}$. However, the bad model performance is caused not only by the altitude difference between the station and the model grid cell average, but also other inhomogeneities, such as strong emission sources in the area. The Schauinsland station, for instance, is located about 10 km from Freiburg city, and about 1 km above it. In the models both the city and the station are covered with one uniformly mixed grid cell, while in reality the winter time low boundary layer traps the pollution below the station altitude. Indications of wintertime overestimation are also visible for some of the other high stations, though with smaller magnitude, as not all the high stations are located at extreme points of the terrain, such as mountain summits, and not all of them have strong emission sources in the immediate vicinity. Opposite problems arise for sites located in narrow valleys, where the models cell-mean altitude is higher than the station and the models overspread the pollution that in reality can be trapped in the valley.

*The stations' locations have been be added to the supplement. The discussion about the station representativity has been extended.*

*5) Section 3.1, page 10, lines 2-4: Explain the reasons for models' different ability in simulating PM2.5 and PM10 in summer and winter.*

For most of the models, temporal correlations and factor-two agreements are better in winter than in summer for both PM fractions, and the spatial correlation of $PM_{2.5}$ is also better in winter. The only score with opposite behaviour is the spatial correlation for $PM_{10}$. The summertime worse scores are probably due to the highly uncertain components that dominate the

summer aerosol - wind-blown dust, wild-land fires, and biogenic secondary organic aerosols. In summer the $PM_{10}$ pattern over Europe is formed by the inflow of Saharan dust and wild-land fires in Portugal and Spain creating a strong south to north gradient. This gradient is reproduced by the models, although with smaller magnitude (Figure A1.1, lower right panel). As LOTOS-EUROS misses the dust and fire contribution, it does not reproduce this pattern. As the species contributing to

5   this summer time south-north gradient are desert dust and wild-land fires, which by nature are episodic and hard to model, the temporal correlation and factor-two agreement are still generally lower and bias is larger in summer. In winter the particulate matter is dominated by the anthropogenic emissions, forming a more complex pattern (Figure A1.1, left panels), and thus the spatial correlation is worse.

*Explanations have been added to the section.*

| | *Winter* | *Summer* |
|---|---|---|
| *$PM_{2.5}$* |
 med djf PM2_5 ug/m3 | med jja PM2_5 ug/m3 |
| *$PM_{10}$* | med djf PM10 ug/m3 | med jja PM10 ug/m3 |

**Figure A1.1. Seasonal average observed and ensemble median PM concentrations for winter and summer.**

*6) Section 3.1, page 10, line 11: The authors state: "The medianComp fully includes SOA, desert dust and fire-induced PM." This is not true since according to Table 2, SOA are not included in LOTOS-EUROS and in SILAM. Also BCs dust*
15   *is not included in LOTOS-EUROS.*

MedianComp is the sum of the ensemble medians of all the individual PM components. Although not all components are provided by all four models, when computing the median field for every component only those models are used which provided a valid field for that component. Thus, medianComp will include the median SOA of EMEP and CMAQ and median dust of SILAM and EMEP. As these models have provided valid fields for these components, the medianComp PM also includes valid fields of SOA and dust.

*The description of MedianComp model has been extended*

*7) Section 3.1, page 10, lines 16-18: The authors explain that the differences between median and medianComp are due to desert dust. However, according to Table 2, it is only the fire originated PM that were not included in total PM while BCs dust was included all CMAQ, EMEP and SILAM.*

That is correct. However, in CMAQ the dust and fire contributions are very low and LOTOS-EUROS does not have them at all, so the median total PM is based on half of the models with zero or very low dust concentration. MedianComp is based only on the valid dust fields of SILAM and EMEP and thus includes noticeably higher dust contribution.

*Explanations have been added.*

*8) Section 3.2, Table 6: Which are the size bins of the PM species validated (comment mostly for sea salt and mineral dust). The words "mineral dust" should be replaces with "nss-Ca".*

- The Ca and Na observations in EMEP network are made mostly in whole aerosol without size limits.

- Dust in the EMAC model is emitted in two log-normal modes with the size distribution parameters from (Dentener et al., 2006): accumulation mode with 0.42 µm median diameter and standard deviation 1.59, and coarse mode with 1.30 µm median diameter and standard deviation 2.00. These modes are selected to cover the whole dust distribution relevant for atmospheric transport (Ginoux et al., 2004, 2001). When the EMAC boundary conditions are projected to the model size bins in EMEP and SILAM, both of the size modes are assumed to be fully inside PM10, which agrees reasonably well with the observations of (Dubovik et al., 2002). The sum of these two modelled modes is used when comparing with the nss-Ca observations.

- For all models the Na concentration for model-measurement comparison was computed from sea salt in PM10. As the models already overestimate Na concentration, if taking into account also the particles larger than 10 µm the real overestimation would be even larger than what is shown. However, comparing Na in PM10 with Na in whole sea salt in SILAM (size range 0.01 to 30 µm), the changes are minor for majority of the EMEP stations that observed Na in 2005: below 5% for 65% of the stations, below 10% for 77%, and below 20% for all stations. The concentration changed more than 10% only in the stations located directly at seaside.

*Table 3 has been updated to include the aerosol sizes for the observations. Reference to the dust size distribution has been added to EMAC model description. Explanations have been added to the results section about the sea salt size distributions.*

*Why evaluation is presented explicitly only for SO2 and not for other PM precursor gas compounds like NH3 or NOx?*

- Observations of NH3 and HNO3 are more prone to artefacts(e.g. Chang et al., 2002; Schaap et al., 2011), thus the less uncertain NH3+NH4 and HNO3+NO3 observations were originally chosen for model evaluation. Those were also available from larger number of stations. The contribution of HNO3 and NH3 to the sum is shown on Figure 4 of the resubmitted manuscript with dark shading. On annual average level the gas phase fraction is in both cases relatively well reproduced by most of the models, only SILAM underestimates HNO3 and overestimates NH3 contributions. However, the models do not reproduce well the seasonal variations in HNO3 and NH3 concentrations – EMEP and LOTOS-EUROS overestimate the seasonal variability of HNO3, CMAQ strongly overestimates the autumn NH3 concentrations and so does with smaller magnitude also SILAM. Temporal correlation coefficients and factor-2 agreements are noticeably worse for NH3 and HNO3 compared with NH4 and NO3 aerosols or the gas-aerosol sums (Table 7 of the resubmitted manuscript). In addition to the uncertainties in the observations, there are other reasons for model errors in these species. Especially for NH3 the timing of the emissions as used in the models (fixed temporal profiles) can deviate substantially from real world emission timing which is largely controlled by meteorology (Backes et al., 2016; Hamaoui-Laguel et al., 2014; Hendriks et al., 2016). Meteorology also influences the total amount of emitted NH3 but the strongest influence is on the timing. The timing of agricultural activities, such as manure spreading has also a strong impact on the emissions (Hendriks et al., 2016). This mismatch may translate into episodes where the models predict high concentrations but due to unfavorable meteorology very little was emitted and vice versa, partly explaining why the models do not properly reproduce the seasonal fluctuations but on annual average are quite OK.
- Regarding NOx, NO was measured by only one EMEP station and thus no thorough evaluation is possible. The model scores and seasonal developments of NO2 concentration are shown on Table A1.1 and Figure A1.2. The seasonal variations of NO2 are well reproduced, but all models apart from CMAQ overestimate NO2 all the seasons. This can be one of the reasons for the overestimation of the sum of NO3- and HNO3.

*NH$_3$ and HNO$_3$ have been added to Figure 4 of the manuscript, model scores for those species have been added to Table 7. Section on the comparison results has been extended. A figure with seasonal variations of NO$_2$ and a table with model scores for it have been added to the supplementary material.*

**Table A1.1. Model skill scores for the gas phase precursors of NO$_3$ and NH$_4$, average over the EMEP stations that observed these species.**

| Species | Model | Scaled bias | tCor | Fac2 |
|---|---|---|---|---|
| **NH$_3$** | CMAQ | 0.04 | 0.18 | 0.25 |
| *Ave obs:* | EMEP | -0.07 | 0.30 | 0.36 |
| *0.75 µg N/m$^3$* | LOTOS-EUROS | 0.19 | 0.22 | 0.38 |
| | SILAM | 0.32 | 0.30 | 0.40 |
| | median | 0.05 | 0.31 | 0.39 |
| **HNO$_3$** | CMAQ | 0.21 | 0.34 | 0.43 |
| *Ave obs:* | EMEP | -0.11 | 0.38 | 0.39 |
| *0.19 µg N/m$^3$* | LOTOS-EUROS | 0.00 | 0.38 | 0.40 |
| | SILAM | -0.53 | 0.32 | 0.32 |
| | median | -0.16 | 0.41 | 0.44 |
| **NO$_2$** | CMAQ | -0.06 | 0.57 | 0.59 |
| *Ave obs:* | EMEP | 0.19 | 0.52 | 0.60 |
| *1.76 µg N/m$^3$* | LOTOS-EUROS | 0.26 | 0.45 | 0.58 |
| | SILAM | 0.39 | 0.54 | 0.57 |
| | median | 0.16 | 0.56 | 0.61 |

[Figure]

**Figure A1.2 Observed and predicted seasonal concentrations of NO$_2$.**

*9) Section 3.2.3: The performance of CMAQ, can it be explained by the fact that fire emissions are included in other PM?*

CMAQ has the largest negative bias and lowest correlation for the carbonaceous aerosols. For EC EMEP is as biased as CMAQ, while being closer to the observations for OC. In CMAQ the fine mode fire emissions were actually included in the primary OA, although not to EC; only the coarse mode fire PM was included in the unspeciated coarse primary PM. For all other models the fire contribution was excluded from comparison with EC and OC. Thus, the missing fire contribution cannot explain the bias in CMAQ and other explanations are needed for the negative bias found for the carbonaceous aerosols.

The fire PM concentrations modelled by EMEP, LOTOS-EUROS and SILAM can be speciated as post-processing following Akagi et al., (2011) or Andreae and Merlet, (2001). On average these papers suggest roughly 5% EC and 50% OC content

for fire emitted aerosol. The fire contribution to EC and OC calculated following this composition is shown on Figure 4 of the resubmitted manuscript with darker shading. SILAM only shows a small fire contribution to EC in spring and summer, while in EMEP and LOTOS-EUROS the contribution is larger and visible all year round. EMEP also predicts a noticeable fire contribution to OC for all seasons. For EMEP and LOTOS-EUROS, the fire contribution reduces the model bias for the carbonaceous species, while at the same time reducing the correlation with the measurements (Table A1.2). The SILAM EC prediction quality does not noticeably change.

*Table 2 has been corrected to show that the fine particles from fires were included in CMAQ as OA. Fire contributions have been added to Figure 6 in the paper. For CMAQ the fire emitted OC is mixed with the primary anthropogenic part and cannot be shown separately and the fire emissions are excluded for EC. The section has been updated so that the carbonaceous aerosols of the other models include the fire contribution.*

**Table A1.2 Comparison of model scores for the carbonaceous species with and without the wildfire contribution**

| Species | Model | *Without firePM* | | | *With firePM* | | |
|---|---|---|---|---|---|---|---|
| | | Scaled bias | tCor | Fac2 | Scaled bias | tCor | Fac2 |
| EC in PM$_{2.5}$ | CMAQ | -0.61 | 0.51 | 0.35 | -0.61 | 0.51 | 0.35 |
| *Ave obs:* | EMEP | -0.60 | 0.56 | 0.43 | -0.56 | 0.53 | 0.4 |
| 1.08 µg C/m$^3$ | LOTOS-EUROS | -0.42 | 0.58 | 0.45 | -0.34 | 0.51 | 0.44 |
| | SILAM | -0.17 | 0.61 | 0.41 | -0.17 | 0.61 | 0.4 |
| | median | -0.51 | 0.61 | 0.37 | -0.45 | 0.6 | 0.38 |
| OC in PM$_{2.5}$ | CMAQ | -0.80 | 0.52 | 0.26 | -0.80 | 0.52 | 0.26 |
| *Ave obs:* | EMEP | -0.38 | 0.58 | 0.64 | -0.25 | 0.54 | 0.6 |
| 3.61 µg C/m$^3$ | median | -0.59 | 0.60 | 0.58 | -0.52 | 0.54 | 0.61 |
| EC in PM$_{10}$ | CMAQ | -0.69 | 0.42 | 0.32 | -0.69 | 0.42 | 0.32 |
| *Ave obs:* | EMEP | -0.70 | 0.43 | 0.37 | -0.66 | 0.46 | 0.35 |
| 1.32 µg C/m$^3$ | LOTOS-EUROS | -0.53 | 0.43 | 0.45 | -0.48 | 0.39 | 0.44 |
| | SILAM | -0.36 | 0.43 | 0.37 | -0.35 | 0.45 | 0.38 |
| | median | -0.61 | 0.46 | 0.38 | -0.58 | 0.49 | 0.37 |
| OC in PM$_{10}$ | CMAQ | -0.85 | 0.36 | 0.18 | -0.85 | 0.36 | 0.18 |
| *Ave obs:* | EMEP | -0.51 | 0.38 | 0.52 | -0.37 | 0.46 | 0.52 |
| 4.78 µg C/m$^3$ | median | -0.67 | 0.40 | 0.45 | -0.61 | 0.46 | 0.48 |

*10) Section 3.2.5: Any comments on OA comparison with observations should be based on CMAQ and EMEP results since the other models include OC in other PM. Similar remark for dust straightforwardly included only in EMEP and SILAM.*

*The text has been changed according to the reviewer's suggestion*

*11) Section 3.2.5., page 15, line 7: Correct OA with OC.*

*Done*

*12) Section 3.2.5., page 15, line 11: Please add that also the OC is for some models included in primary aerosols (as the case for LOTOS-EUROS and SILAM)*

*Done*

*13) Section 3.2.5., Figure 8: a) suggestion: add the simulated SOA and POA and compare them with the observed TOA, b) why fire PM are presented separately and not speciated in OC and EC so as to allow better comparison with observations? (see also comment 1). In the legend of Figure 8 the PM species are not presented in the same order as they appear in the plots.*

10    a) As total OC is measured the observations on Figure 8 are shown as total OA. The modelled OA has been shown as POA and SOA separately, to demonstrate how the contribution of these varies between the models and the stations.

b) Apart from CMAQ that included the fire emissions in OA, the fire emissions were modelled as unspeciated particulates. The concentration can be speciated following the average composition from Akagi et al., (2011) and Andreae and Merlet, (2001), giving ~90% OM, ~5% EC and ~5% non-carbonaceous compounds.

15    *Figure 8 has been updated to show the fire contribution to OC and EC. The legend has been fixed.*

*14) Section 4.1: This section provides a theoretical description of the possible reasons for overestimations or underestimations in the model results. However, there is no quantification of the uncertainties. For example, how much is the uncertainty introduced in the model validation because of the assumption that dust and anthropogenic mineral*

*aerosols have a 10% and 3.5 Ca content respectively? Clarifications also are necessary in the last paragraph of this section in which the authors discuss about the absence of fire emissions from the computations while in Table 2 fire originated PM are presented as included in the simulations of all models.*

As the models did not include any other dust sources than the inflow from the boundaries, underestimation of dust concentration is expected and comparison with nss-Ca confirms it, while not providing its exact magnitude.  Various sources

25    give Saharan dust Ca content from <5% to >15% and for the anthropogenic emissions the variations are even larger, depending on the source sector and local soil, so the uncertainty in aerosol Ca content can be expected to be a few times. However, with the 10 and 3.5% Ca content, the EMEP model underestimated the nss-Ca by 75% and SILAM by 58%, so even assuming twice the calcium content, the nss-calcium concentrations would still be underestimated by the models. This uncertainty does not influence the accuracy of the predictions and evaluation of $PM_{10}$ and $PM_{2.5,}$ as primary PM and dust

30    were modelled as totals and the 3.5% and 10% factors were only applied when comparing with the nss-Ca observations.

In EMEP and LOTOS-EUROS, using the IS4FIRES v1 emissions resulted in degradation of model scores for $PM_{2.5}$ and $PM_{10}$ and thus these models excluded the fire PM from their total $PM_{2.5}$ and $PM_{10}$ fields, while still providing it as a separate field. In SILAM, a newer version of the emission data was used (IS4FIRES v2, (Soares et al., 2015), together with dynamic emission vertical profiles of (Sofiev et al., 2012), while in other models,  the IS4FIRES v1 emission data was spread evenly

to the first 1000m. Mainly due to the vertical profiles that release most of the smoke high aloft, the ground level concentrations of fire PM were substantially lower in SILAM and thus the fire PM does not significantly affect the model performance for PM, demonstrating that the quality of the fire emission data is essential for simulating the particulate matter concentrations.

5    *Clarifications have been added.*

*15) Section 4.3: It is a very interesting section. The authors should present results on the models' improved performance when water is accounted for (even in the four stations providing a complete set of PM measurements; after all section 3.2.5 was based on the measurements in these few stations).*

Based on the dry PM mass and speciation provided by the models, aerosol thermodynamic model ISORROPIA2 (Fountoukis
10   and Nenes, 2007) was applied to evaluate the equilibrium water content at the conditions where the filters were weighted (20°C, 50% relative humidity). ISORROPIA2 was run in the reverse mode, where the input quantities were the soluble inorganic components (SIA, sea salt, Ca) in the aerosol phase. The aerosols were assumed to be internally mixed. Both stable and metastable states were computed, corresponding to the lower and upper branches of the deliquescence hysteresis loop, the latter one describing the case when the aerosol has been exposed to more humid environment and crystallization has not
15   occurred. In this way lower and upper limits of the aerosol bound water amount can be estimated.

ISORROPIA2 was applied to estimate the water contribution at all the EMEP sites that measured $PM_{2.5}$ or $PM_{10}$. In the stable case, the annual mean $PM_{2.5}$ water content, average over all EMEP stations, stayed between 4 and 9% depending on the model, and between 11 and 17% for $PM_{10}$. For $PM_{2.5}$, the models predicted annual average water content above 10% for only a few stations. For $PM_{10}$, CMAQ and EMEP predict majority of the stations to have less than 10% of water content, while
20   LOTOS-EUROS and SILAM predict the majority to be between 10 and 20%; annual average water contents of more than 25% were predicted for some stations. The water content of $PM_{10}$ computed in the metastable mode was on average about twice higher (~25%); ~20% water content was predicted for $PM_{2.5}$.

As seen from Table 6 of the resubmitted manuscript, adding the aerosol-bound water reduces noticeably the model bias for both $PM_{10}$ and $PM_{2.5}$. For $PM_{2.5}$ the correlation coefficients are not much affected, while for $PM_{10}$, they are noticeably
25   reduced. The factor-2 agreements improve due to the bias reduction. The worse correlations could be related to the models overestimating the sea salt concentrations that can lead to overestimation of the water content in $PM_{10}$, as it is the most hydrophilic of the considered aerosol components.

Also other uncertainties exist estimating of the PM water content. Firstly, ISORROPIA2 computes the water content based on the inorganic part of aerosol – SIA, sea salt, calcium; it does not take into account the water related to the hydrophilic part
30   of the organic aerosol. Secondly, the aerosols were assumed fully internally mixed, which lowers the deliquescence humidity compared to external mixtures and might lead to overestimation of water uptake.

The water contribution for the four stations with most complete aerosol composition observations can be seen on Figure 8 of the resubmitted manuscript. Adding the aerosol bound water in metastable state closes the gap between the observed total $PM_{10}$ and the sum of the individual components in all stations (in Montseny the $PM_{10}$ estimate based on nearby stations can

be inaccurate). In Ispra and Birkenes the observed PM is exceeded, which could indicate that the aerosol on the filters is in crystallized state or be due to inaccuracies in other observed species. In Ispra, errors were suspected in the observations of the carbonaceous components, while in Birkenes, the sea salt observations are taken from all aerosol, not $PM_{10}$. The models predict very high aerosol water content for Birkenes and noticeably overestimate the $PM_{10}$ there, as the overestimated sea salt concentration leads to very high water uptake of the aerosol.

*The water contribution has been added to figures 3 and 8 and Table 6 has been divided to two parts, the first of those comparing the model scores for the dry PM to the water equilibrated PM. The discussion regarding the aerosol-bound water has been extended according to the reviewers suggestion.*

*16) Figure 6: Correct in the lower-right plot the OC to EC.*
*Done*

*17) Figure S6: Was CMAQ excluded from the average since dust in CMAQ was included in other PM?*
Yes, the figures are based only on the models that included the shown component explicitly.
*The figure captions have been updated to include this information.*

*18) Figure S7: Was LOTOS-EUROS and SILAM excluded from OA mean values since OA in these modes were included in other PM? The Figure is not commented in the text of the manuscript.*

The OA figure is now based on EMEP model only, as SILAM and LOTOS-EUROS included primary OA in PM and CMAQ included in its OA the fire emissions.
*The figure caption has been updated to include this information.*

*19) Figure S8: Please check comment 1. How fire emissions were simulated and consequently presented in the Figure for PM? If speciated to OC and EC emissions, how fire-related OC and EC concentrations were distinguished from anthropogenic OC and EC concentrations?*
CMAQ emitted OC, which cannot be distinguished from the anthropogenic OC. EMEP, LOTOS-EUROS and SILAM emitted unspeciated $PM_{2.5}$ and $PM_{10}$ and transported them as separate fields. Figure S8 shows the median of those fields.
*The caption of Figure S8 has been updated*

5  The paper of Curci et al presents a detailed study on nitrate formation in Po Valley and how its concentration is influenced by the boundary layer processes and temperature and humidity vertical profiles. It is indeed an interesting question, how well these effects are taken into account in the regional models. The models use a range of different algorithms for SIA creation, such as thermodynamic equilibrium model ISORROPIA2 in LOTOS-EUROS or separate equilibrium computations for $NH_4NO_3$ and a parameterization for coarse $NO_3$ production on sea salt particles in SILAM and EMEP. Generally these

10  should be capable of simulating the profile of $NO_3$ creation and brake up, provided that the model vertical resolves the temperature and relative humidity profiles. However, this can easily not be the case - for instance LOTOS-EUROS has a single layer representing the whole boundary layer and SILAM layers at boundary layer top can easily be 500-1000 meters thick.

While the low vertical resolution of the gathered model data does not allow for a detailed analysis of this effect in the whole

15  dataset, elevated plumes of $NH_4NO_3$ are clearly visible in SILAM model output (**Error! Reference source not found.**), mainly because the efficient dry deposition of $HNO_3$ depletes its concentration in the near-surface layers making $NH_4NO_3$ break to the gaseous compounds. Thus, the formation of the residual layers and mixing them down in the morning and the ability of the models to reproduce these effects can have an impact on the simulation quality. However, as the EMEP aerosol composition observations have daily resolution, it's hard to tell, how much these processes influence the model-

20  measurement correlation compared with all the other uncertainties related to $NH_4NO_3$ formation, such as the temperature dependence of $NH_3$ emission.

*The discussion of the uncertainties in $NO_3$ modelling has been extended and the reference added to the manuscript.*

[Figure]

**Figure A2.1 Summer time concentration profiles of NH4NO3 from SILAM model, west-east and south-north cross-sections of Europe, selected to cut through Po Valley.**

5    *Could this be shown, or the results commented on the NO3+HNO3 concentration that the reader may actually directly see in the plots?*

The reason for selecting $HNO_3+NO_3$ as the presented variable was, that there are more stations in EMEP network that measure the sum than there are those that measure $HNO_3$ alone, also the sum is measured with higher accuracy, while the HNO3 concentration is more prone to observation artefacts .

10   Figure 4 in the resubmitted manuscript shows $HNO_3+NO_3$ together with the contribution from $HNO_3$ (shaded). On annual average level all models overestimate the sum by 5-35 %. EMEP and LOTOS-EUROS accurately predict the $HNO_3$ fraction in the sum, while SILAM underestimates and CMAQ overestimates it by 10%. However, the models do not reproduce well the seasonal variations in $HNO_3$ concentration.

*Figure 4 and related discussion has been updated in the paper.*

*- page 9, line 10: it is not completely true that the seasonal cycle is not reproduced by all SIA, e.g. NO3 and NH4 are reproduced quite well.*

The sentence has been restated.

5 *- page 9, line 16: "... overestimate the temperature dependence ..." suggest to rephrase with "... have an exaggerated temperature dependece ..." to avoid confusion with overestimated/underestimated resulting PM values*

*Done*

*. - page 9: in general, natural PM seems to be a major factor contributing to the spring PM maximum: may you confirm*

10 *that (or not)?*

Of the natural aerosols considered (Figure 5 in the manuscript), sea salt exhibits no spring peak - elevated $Na^+$ concentrations are observed only in winter and the models reproduce this behaviour. Dust, on the other hand, seems to be contributing – observed nss-Ca concentrations peak in spring and so do the modelled Saharan dust concentrations. Previous studies about Saharan dust have found the emissions peaking at spring (Fiedler et al., 2013; Laurent et al., 2008). Additionally to desert

15   dust there are other possible reasons for elevated crustal aerosol concentration in spring, such as agricultural activities and vehicle caused erosion of roads - in colder regions where winter tires are used and the roads are sanded against slipperiness high dust emissions occur when the road conditions get dry in spring.

Considering the biogenic primary particles, pollens are abundant in air in spring - maximum concentrations can reach a few tens of thousands of pollen/$m^3$, equivalent to tens of µg/$m^3$. However, their size is mostly too large to make them relevant for

20   $PM_{10}$ (~20 µm for abundant early spring flowering species like birch). Evidence exist, that pollen grains can break in atmosphere and produce particles in size range relevant for even $PM_{2.5}$, however, such particles are unlikely to be abundant enough to be noticeable in the PM budget. Also the fungal spores become abundant only in summer – high concentrations can be observed from June to October.

Regarding the anthropogenic contribution, all observed secondary inorganic species show spring maxima.

*- page 10, lines 16-17: perhaps could be useful the discussion on EC lifetimes presented in this paper: Wang, X., Heald, C. L., Ridley, D. A., Schwarz, J. P., Spackman, J. R., Perring, A. E., Coe, H., Liu, D., and Clarke, A. D.: Exploiting simultaneous observational constraints on mass and absorption to estimate the global direct radiative forcing of black carbon and brown carbon, Atmos. Chem. Phys., 14, 10989-11010, doi:10.5194/acp- 14-10989-2014, 2014.*

30   The paper of Wang et al reports shorter EC lifetimes to be more consistent with observations and a similar conclusion was reached by Samset et al., (2014) - both studies found that shorter lifetime was 
[revised manuscript text omitted]